# People are more error-prone after committing an error

Tyler J. Adkins [1], Han Zhang [1] & Taraz G. Lee [1] ✉

Humans tend to slow down after making an error. A longstanding account of this post-error slowing is that people are simply more cautious. However, accuracy typically does not improve following an error, leading some researchers to suggest that an initial 'orienting' response may initially impair performance immediately following error. Unfortunately, characterizing the nature of this error-based impairment remains a challenge in standard tasks that use free response times. By exerting control over the timing of responses, we reveal the time course of stimulus-response processing. Participants are less accurate after an error even when given ample time to make a response. A computational model of response preparation rules out the possibility that errors lead to slower cognitive processing. Instead, we find that the efficacy of cognitive processing in producing an intended response is impaired following errors. Following an error, participants commit more slips of action that tend to be a repetition of the previous mistake. Rather than a strategic shift along a single speed-accuracy tradeoff function, post-error slowing observed in free response time tasks may be an adaptive response to impaired cognitive processing that reflects an altered relationship between the speed and accuracy of responses.

A longstanding finding in the field of psychology is that people respond more slowly immediately following errors in decision-making[1,2]. This phenomenon is often referred to as "post-error slowing"[3]. Post-error slowing often coincides with increased response accuracy and has thus been widely assumed to reflect an adaptive strategic adjustment to prevent future errors. The initial prominent accounts of this phenomenon suggested that people are more cautious in responding after they make an error. Neural network models explain these post-error effects in terms of a decrease in baseline activation of a response[4], and evidence accumulation models explain these post-error effects in terms of an increase in the threshold of evidence required to make a response[5]. This would all predict that we should be more accurate after making an error as we shift along the same speed-accuracy tradeoff curve. However, accuracy is quite often stable or even reduced after an error, especially when the interval between each response and the subsequently presented stimulus is short[2,6]. This has led some researchers to instead conclude that post-error slowing is a maladaptive response

reflecting impaired processing rather than a cognitive control adjustment aimed at improving behavior. Some researchers have attempted to reconcile adaptive and maladaptive accounts of post-error slowing[7–11]. In several of these accounts, following an error, there is an initial transient and reflexive 'orienting' response toward the source of the error that can impair or otherwise inhibit cognitive processing. This is followed by a subsequent error-specific strategic response that can aid performance. In a recent prominent account, Wessel[10] proposed an adaptive orienting theory whereby an unexpected action outcome triggers an initial automatic global inhibition of motor and cognitive processes that is followed by error-specific adaptive control. Unfortunately, characterizing the nature of impairments in cognitive processing underlying post-error responses has remained a challenge in standard paradigms that use free response times given the known challenges with evaluating speed-accuracy tradeoffs[12]. It is currently unknown exactly why and for how long accuracy is often so poor following errors, given how much slower people are to respond.

[1]Department of Psychology, University of Michigan, Ann Arbor, MI, USA. ✉e-mail: tarazlee@umich.edu

Post-error effects are usually examined in tasks that measure free response times (RT) and error rates. A critical issue with free RTs is that they confound the cognitive processing necessary for response preparation (e.g., stimulus identification and action selection) with response initiation (i.e., emitting the motor response). Indeed, recent work has shown that responses are accurately prepared and ready to deploy much more quickly than free RT would indicate[13]. Furthermore, RT can be habitually set by prior experience and does not solely reflect the time required for the computations necessary for the task at hand[14]. This collection of work argues that preparation and response initiation are independent motor control parameters[13–15]. On this view, people not only decide *what* response to make but also *when* to make it. Prior studies on post-error effects do not directly distinguish between selection and initiation in their experimental designs or in their theoretical models. This presents a problem in that slower RTs after errors could be due to strategic delays in response initiation without any change in the cognitive processes underlying response preparation per se. Although researchers have used mathematical models such as the drift-diffusion model to attempt to tease apart response caution from maladaptive cognitive processing following an error[5,8], this approach fundamentally relies on fitting RT distributions with the assumption that RT is a reliable indicator of the total duration of cognitive processing necessary to produce a response. (Note that although *non-decision time* is an estimated parameter in the drift-diffusion model that can influence RT, it is not thought to reflect the cognitive processing that leads to decision). Additionally, the drift-diffusion framework makes fairly strong assumptions about the nature of the decision-making process (a single evidence accumulation process, a static evidence accumulation rate per condition, etc.).

To address these shortcomings, we examined post-error effects using a forced-response paradigm which controls response time and treats processing time as an independent variable[13,15]. In this paradigm (Fig. 1A), participants are cued to respond at the same time on each trial while the onset of the target stimulus is uniformly varied in a 4-alternative forced choice stimulus–response task (Fig. 1B). This allows us to examine post-error effects on processing per se by controlling the time of response initiation to query the state of cognitive processing as time unfolds. We use these data to fit a model that makes minimal assumptions about the cognitive processes leading to a response: (1) each stimulus leads to the preparation of the appropriate response with some mean latency and normally distributed trial-to-trial variability; (2) if stimulus-based response preparation is not yet complete, participants will guess randomly; and (3) action selection following perceptual processing is not perfect and "slips of action"

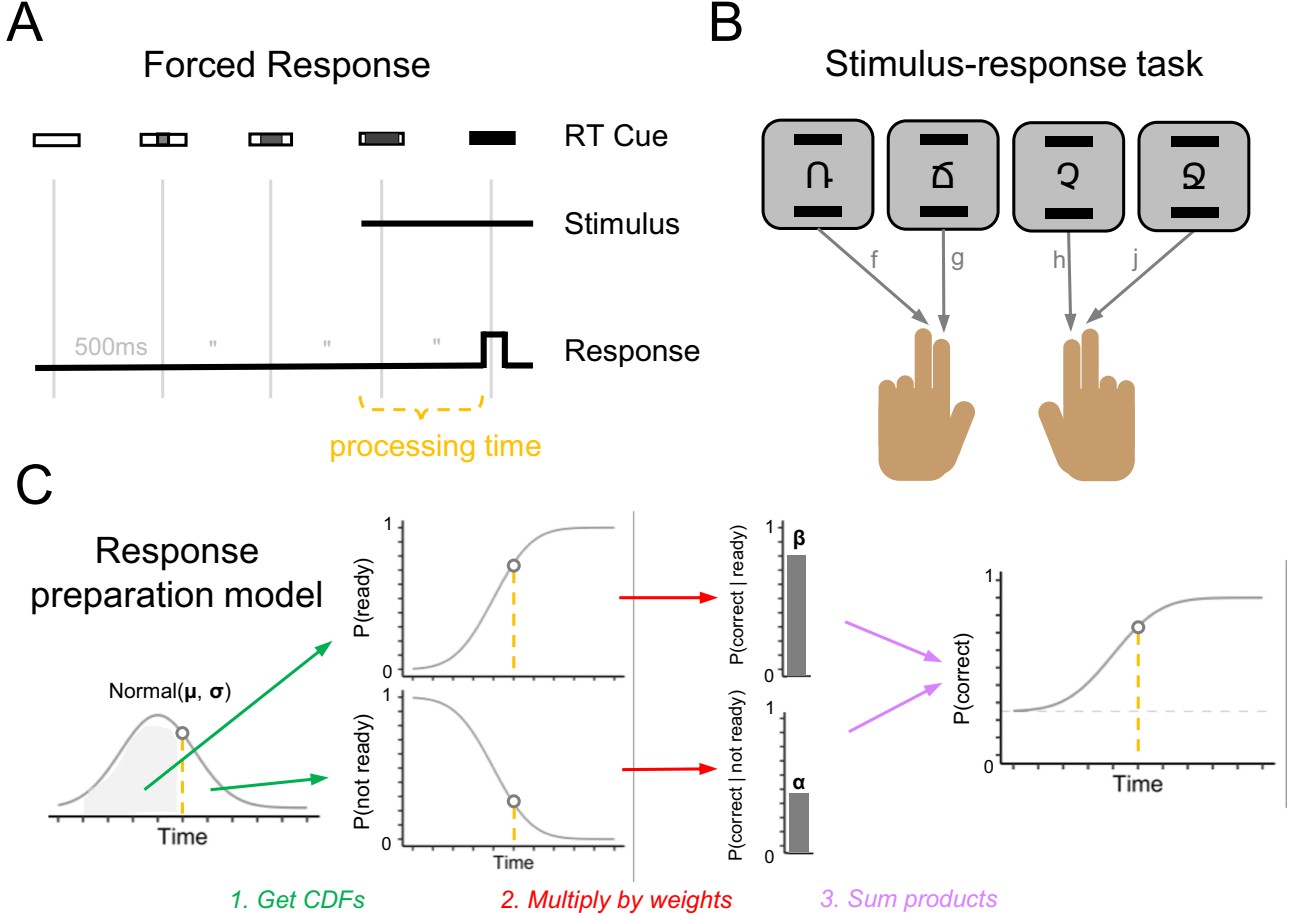

**Fig. 1 | Forced response SR task.** Participants performed a stimulus–response task (4AFC) with forced responding. **A** Participants were trained and cued to respond when an empty rectangle was completely filled (at 2000 ms), and the target stimulus appeared at a random time between 0 and 2000 ms. **B** Participants were instructed to press the 'f' key with their left middle finger, the 'g' key with their left index finger, the 'h' key with their right index finger, or the 'j' key with the left middle finger, depending on the symbol (Experiments 1 and 2). **C** Response preparation model used to predict participants responses as a function of the time available for processing (orange dashed line). The model assumes that the time at which the participant has processed the stimulus and prepared the response is normally distributed. The probability that a response is prepared (or not) at a given time is determined by the cumulative distribution function (CDF) of the Normal ($\mu$, $\sigma$). These probabilities are multiplied by weights representing the probability of expressing the correct response given that it is prepared ($\beta$) or not ($\alpha$). Summing these products gives the probability of expressing the correct response at a given time.

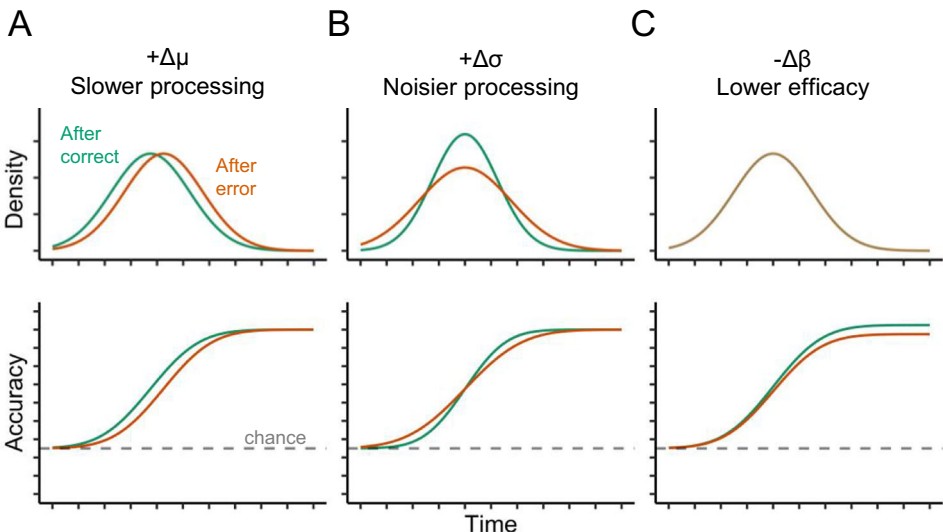

**Fig. 2 | Hypothetical effects of previous error on response preparation.** The first row depicts the effects in terms of the underlying response preparation distributions posited by the response preparation model used in the present study. The second row depicts the effects in terms of behavioral accuracy as a function of preparation time. Each column represents the effect of past errors on a unique parameter in the model. Column **A** depicts slower response preparation after an error, Column **B** depicts noisier response preparation after an error, and Column **C** depicts reduced efficacy of prepared responses after an error. Efficacy is not a property of the underlying distributions but is instead a probability weight assigned to the prepared response—i.e., how likely is the participant to overtly express a response if it has already been prepared. A decrease in efficacy can be understood as an increase in the propensity for an action slip. Note that these parametric changes are associated with distinct predictions about the observable conditional accuracy functions (Row 2). (Note that these conditional accuracy functions are simulated data generated by allowing only one parameter to vary in our model at a time).

sometimes occur[16]. These assumptions lead directly to four free parameters in this model: $\alpha$, the likelihood of a correct response given no response has been prepared (i.e., guessing); $\mu$, the average speed of cognitive processing underlying a correct response; $\sigma$, the standard deviation (i.e., trial-to-trial variability) of the speed of cognitive processing; and $\beta$, the probability a correct response will be produced when this cognitive processing is complete (i.e., the "efficacy" of stimulus-based action selection). $1 - \beta$ is the probability that an action slip occurs even if it is very likely that enough time has elapsed for stimulus-based action selection to occur. (Note that the model is agnostic as to whether action slips are due to problems at the time of response initiation or during response preparation itself.) When combined, these parameters can be used to predict accuracy when the amount of time given for stimulus–response processing is known (Fig. 1C; see Methods for complete modeling details). In combination with controlling the time of response initiation, this modeling framework allows us to distinguish among several distinct ways in which cognitive processing might be affected following an error. Following an error, stimulus–response processing might be slower (Fig. 2A), more variable (Fig. 2B), or simply less effective at producing the correct response resulting in more frequent slips of action (Fig. 2C). Each of these possibilities would map onto a quantitative change in just one of the parameters of our model. As seen in the bottom row of Fig. 2, simulated data from our model shows that each parameter uniquely controls a certain aspect of a participant's speed-accuracy tradeoff function (e.g., slower processing coinciding with a change in $\mu$ would shift the psychometric function to the right without affecting its slope or asymptote).

Across four experiments (Fig. 1B), we show that accuracy is reduced after errors, even when there is ample time to prepare a response (up to 2 s). Our modeling results reveal that this effect cannot be explained by slowed or more variable cognitive processing after an error (i.e., more time required to select a response or trial-to-trial variability in this processing time). Instead, the observed behavior is due to a decrease in efficacy, or the probability of executing prepared responses. In other words, even when the stimulus timing suggests that a correct response is highly likely to be prepared, participants commit more perseverative slips of action following an error. These results suggest that the increased response caution observed in prior studies examining post-error slowing does not simply reflect a shift along the speed-accuracy curve but may rather be an adaptive response to impairments in cognitive processing following an error that coincides with an altered relationship between speed and accuracy.

## Results
### Experiment 1
Following a brief training session to familiarize participants with stimulus–response mappings and the response timing required, participants completed 400 trials of four alternative forced-choice stimulus–response tasks (see Fig. 1 and Methods for complete details). Participants were required to make their responses between 1900 and 2100 ms following the start of each trial, but the stimulus presentation varied randomly between 0 and 2000 ms. This approach allowed us to investigate participants' accuracy while tightly controlling the amount of time available for the cognitive processing required to translate the stimulus information into a response.

Overall, there was strong evidence that participants were moderately less likely to perform the correct response if they made an error on the previous trial [regression coefficient ($b$) = −0.28, 95% credible interval (CI) = [−0.41, −0.15], probability of direction (pd) = 1.0]. However, a sliding window analysis revealed that the effect of previous error on accuracy depended on the amount of time available for preparation (Fig. 3a). For earlier processing times (PT < 500 ms), there was no evidence for an effect of past error on future accuracy ($b$ = 0.06, CI = [−0.25, 0.35], pd = 0.68). For later processing times (PT > 1000 ms), there was strong evidence that participants were much less likely to make the correct response following an error on the previous trial ($b$ = −0.63, CI = [−0.98, −0.25], pd = 1.0).

We next characterized the observed conditional accuracy functions (Fig. 3a) using a mathematical model of the underlying response preparation processes (Fig. 1C). Specifically, we used hierarchical

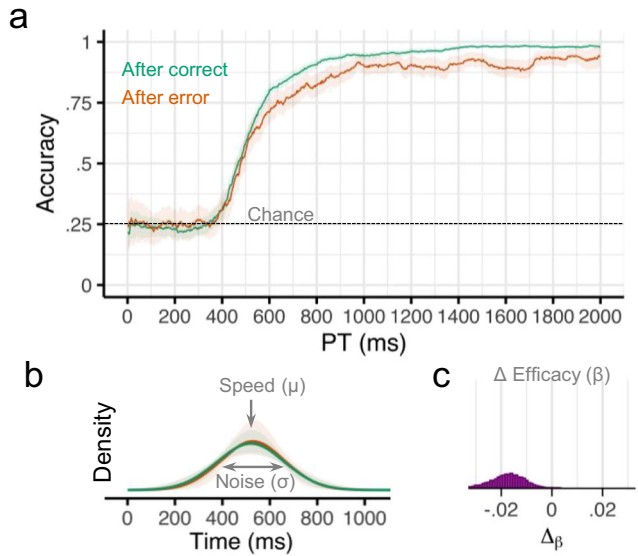

**Fig. 3 | Results from Experiment 1. a** Smoothed response accuracy as a function of preparation time and previous error. Bold lines represent smoothed means, and ribbons represent smoothed 95% confidence intervals (i.e., standard error times 1.96). Note in the response preparation model, upper bound accuracy is controlled by the efficacy parameter $\beta$. **b** Model-estimated probability densities representing the time required to prepare responses following correct and incorrect trials. Densities were computed using group-level intercepts and slopes for $\mu$ and $\sigma$. Bold lines represent the posterior medians, and ribbons represent the 95% quantile intervals of the posterior. **c** Posterior (MCMC) distribution for group-level effects of previous error ($\Delta$) on efficacy $\beta$. $n = 46$ human participants.

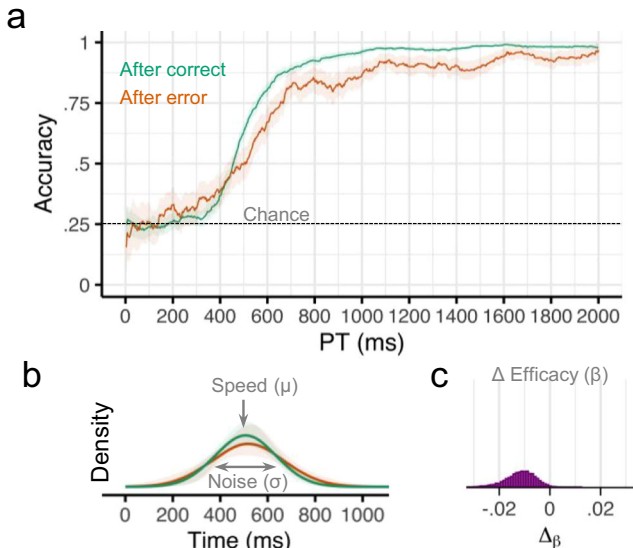

**Fig. 4 | Results from Experiment 2. a** Smoothed response accuracy as a function of preparation time and previous error. Bold lines represent smoothed means, and ribbons represent smoothed 95% confidence intervals (i.e., standard error times 1.96). Note in the response preparation model, upper bound accuracy is controlled by the efficacy parameter $\beta$. **b** Model-estimated probability densities representing the time required to prepare responses following correct and incorrect trials. Densities were computed using group-level intercepts and slopes for $\mu$ and $\sigma$. Bold lines represent the posterior medians, and ribbons represent the 95% quantile intervals of the posterior. **c** Posterior (MCMC) distribution for group-level effects of previous error ($\Delta$) on efficacy $\beta$. $n = 33$ human participants.

Bayesian analyses to fit our response preparation model to all participants' data simultaneously. This analysis returns a posterior distribution over the value of each parameter in our model. Here we report the median of the posterior distribution (M) and the 95% credible interval (CI). This model provided very good fits to our participants' data across all four experiments (see Supplementary Figs. 1 and Methods).

Overall, the expected cognitive processing time required to prepare a response ($\mu$) in this task was 523 ms (CI = [502, 544]), the variability in preparation time ($\sigma$) was 140 ms (CI = [109, 175]), and the efficacy of the prepared response ($\beta$) was 0.97 (CI = [0.96, 0.98]). Efficacy was slightly lower after an error (M = 0.96, CI = [0.94, 0.97]) compared to after no error (M = 0.97, CI = [0.96, 0.98]. The $\Delta_{\beta}$ parameter was negative for nearly all posterior samples, indicating that efficacy post-error was consistently lower than efficacy following a correct trial (M$_{diff}$ = −0.02, CI = [−0.03, −0.01], pd = 0.999; Fig. 3c). There was no evidence that preparation speed was affected by previous errors (M$_{diff}$ = −10 ms, CI = [−40, 50], pd = 0.68) or that preparation variability was affected by previous errors (M$_{diff}$ = −10 ms, CI = [−60, 50], pd = 0.59; Fig. 3b).

## Experiment 2
We conducted an exact replication of the first experiment to test the reproducibility of the above finding that errors reduce efficacy on subsequent trials. There was some evidence that participants were slightly less likely to perform the correct response if they made an error in the previous trial (b = −0.11, 95% CI = [−0.24, 0.04], pd = 0.93). However, a sliding window analysis revealed that the effect of the previous error on accuracy depended on the amount of time available for preparation (Fig. 3a). For later processing times (PT > 1000 ms), there was strong evidence that participants were less likely to perform the correct response following an error on the previous trial (b = −0.50, CI = [−0.93, −0.05], pd = 0.99). For earlier processing times

(PT < 500 ms), accuracy was closer to chance levels, and there was only some evidence that accuracy was higher after an error (b = 0.17, CI = [−0.08, 0.42], pd = 0.91).

Next, we accounted for the observed conditional accuracy functions (Fig. 4a) using a mathematical model of the underlying response preparation processes. Overall, the expected cognitive processing time required to prepare a response (speed, $\mu$) was 512 ms (CI = [481, 542]), the variability in preparation time (variability, $\sigma$) was 141 ms (CI = [108, 178]), and the probability of expressing a response if it is prepared (efficacy, $\beta$) was 0.97 (CI = [0.96, 0.98]). Efficacy was slightly lower after an error (M = 0.97, CI = [0.95, 0.98]) compared to after no error (M = 0.98, CI = [0.97, 0.99]. Although the credible intervals overlapped, the $\Delta_{\beta}$ parameter was negative for nearly all posterior (MCMC) samples, indicating that efficacy post-error was consistently lower than efficacy post-correct (M$_{diff}$ = −0.01, CI = [−0.02, −0.002], pd = 0.99; Fig. 3c). There was little evidence that preparation speed was affected by previous errors (M$_{diff}$ = 10 ms, CI = [−20, 50], pd = 0.71) or that preparation variability was affected by previous errors (M$_{diff}$ = 30 ms, CI = [−20 ms, 70 ms], pd = 0.84; Fig. 3b). These results replicate the finding from Experiment 1 that errors impair subsequent cognitive processing by reducing the efficacy of a prepared response, thereby leading to an increase in slips of action.

## Experiment 3
One possibility for why we did not observe post-error slowing of cognitive processing in the first two experiments is because the time between trials was too long. Previous research has suggested that post-error slowing is reduced as the duration between a stimulus and a previous response grows[7,17]. We, therefore, conducted a pair of replication studies to assess whether the post-error effects observed previously depended on the duration of time between trials. Whereas in the first two experiments, the inter-trial-interval (ITI) was 1000 ms, in the present Experiment 3, the ITI was 0 ms. Overall, there was again

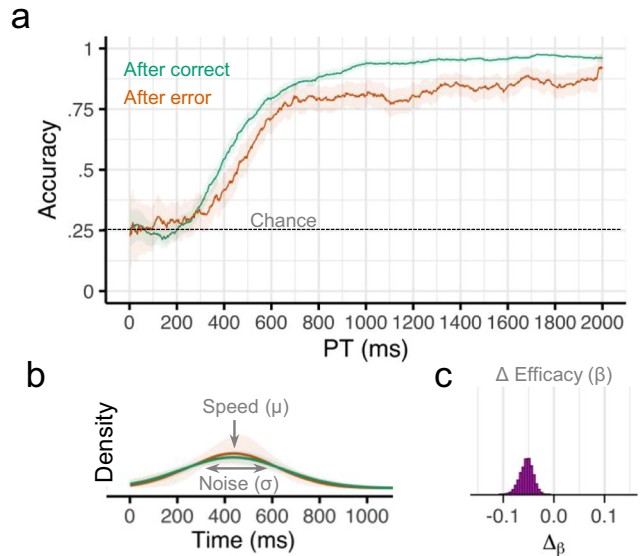

**Fig. 5 | Results from Experiment 3. a** Smoothed response accuracy as a function of preparation time and previous error. Bold lines represent smoothed means, and ribbons represent smoothed 95% confidence intervals (i.e., standard error times 1.96). Note in the response preparation model, upper bound accuracy is controlled by the efficacy parameter $\beta$. **b** Model-estimated probability densities representing the time required to prepare responses following correct and incorrect trials. Densities were computed using group-level intercepts and slopes for $\mu$ and $\sigma$. Bold lines represent the posterior medians, and ribbons represent the 95% quantile intervals of the posterior. **c** Posterior (MCMC) distribution for group-level effects of previous error ($\Delta$) on efficacy $\beta$. $n = 47$ human participants.

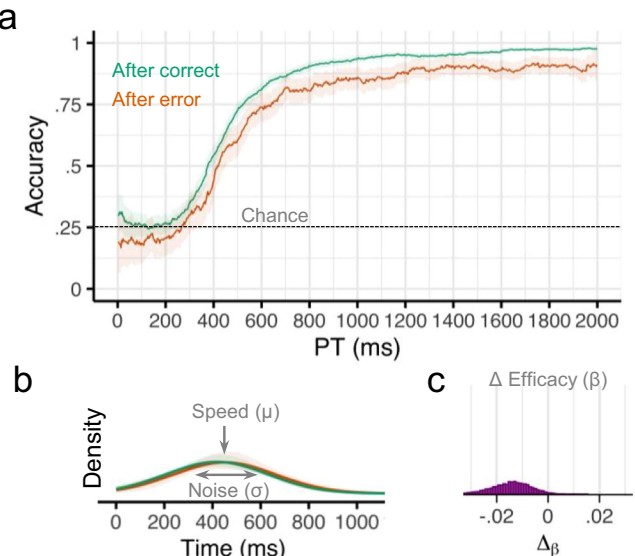

**Fig. 6 | Results from Experiment 4. a** Smoothed response accuracy as a function of preparation time and previous error. Bold lines represent smoothed means, and ribbons represent smoothed 95% confidence intervals (i.e., standard error times 1.96). Note in the response preparation model, upper bound accuracy is controlled by the efficacy parameter $\beta$. **b** Model-estimated probability densities representing the time required to prepare responses following correct and incorrect trials. Densities were computed using group-level intercepts and slopes for $\mu$ and $\sigma$. Bold lines represent the posterior medians, and ribbons represent the 95% quantile intervals of the posterior. **c** Posterior (MCMC) distribution for group-level effects of previous error ($\Delta$) on efficacy $\beta$. $n = 46$ human participants.

strong evidence that participants were less likely to perform the correct response if they made an error on the previous trial ($b = -0.42$, 95% CI = [−0.62, −0.22], pd = 1.0). However, a sliding window analysis revealed that the effect of previous error on accuracy depended on the amount of time available for preparation (Fig. 5a). For earlier processing times (PT < 500 ms), there was little evidence for an effect of past error on future accuracy ($b = -0.14$, CI = [−0.44, 0.15], pd = 0.82). For later processing times (PT > 1000 ms), there was strong evidence that participants were much less likely to perform the correct response following an error in the previous trial ($b = -0.87$, CI = [−1.20, −0.49], pd = 1.0).

Again, we characterize the observed conditional accuracy functions (Fig. 5a) using a mathematical model of the underlying response preparation processes (Figs. 1C and 2). Overall, the expected cognitive processing time required to prepare a response ($\mu$) in this task was 434 ms (CI = [385, 479]), the variability in preparation time ($\sigma$) was 202 ms (CI = [155, 265]), and the efficacy of the prepared response ($\beta$) was 0.94 (CI = [0.92, 0.96]). Efficacy was slightly lower after an error (M = 0.91, CI = [0.87, 0.94]) compared to after no error (M = 0.96, CI = [0.95, 0.97]). The $\Delta_\beta$ parameter was negative for all posterior samples, indicating that efficacy post-error was consistently lower than efficacy post-correct ($M_{diff} = -0.05$, CI = [−0.08, −0.03], pd = 1.0; Fig. 5c). There was no evidence that preparation speed was affected by previous errors ($M_{diff} = 0$ ms, CI = [−50 ms, 50 ms], pd = 0.50; Fig. 5b) or that preparation variability was affected by previous errors ($M_{diff} = -20$ ms, CI = [−90 ms, 60 ms], pd = 0.73; Fig. 5b).

## Experiment 4
In this experiment, we imposed an inter-trial interval (ITI) of 2000 ms. If the post-error effects reported above were due to an orienting response to an unexpected event[6] or transient distraction (i.e., flustering), then these effects might disappear during a sufficiently long ITI —however, this was not the case. Overall, there was strong evidence

that participants were less likely to perform the correct response if they made an error in the previous trial (b = −0.23, 95% CI = [−0.43, −0.04], pd = 0.99). A sliding window analysis revealed that the effect of previous error on accuracy was relatively constant across processing times (Fig. 6a). There was evidence that participants were less likely to perform the correct response following an error on the previous trial for earlier processing times (PT < 500 ms; $b = -0.42$, CI = [−0.78, −0.10], pd = 0.995) as well as for later processing times (PT > 1000 ms; $b = -0.34$, CI = [−0.67, 0.04], pd = 0.97).

Again, we characterize the observed conditional accuracy functions (Fig. 6a) using a mathematical model of the underlying response preparation processes (Figs. 1c and 2). Overall, the expected cognitive processing time required to prepare a response ($\mu$) in this task was 434 ms (CI = [396, 471]), the variability in preparation time ($\sigma$) was 214 ms (CI = [175, 257]), and the efficacy of the prepared response ($\beta$) was 0.96 (CI = [0.94, 0.97]). Efficacy was slightly lower after an error (M = 0.95, CI = [0.93, 0.97]) compared to after no error (M = 0.97, CI = [0.95, 0.98]). The $\Delta_\beta$ parameter was negative for nearly all posterior samples, indicating that efficacy post-error was consistently lower than efficacy post-correct ($M_{diff} = -0.01$, CI = [−0.03, −0.00], pd = 0.98; Fig. 6c). There was some evidence that preparation speed was slower after an error ($M_{diff} = 30$ ms, CI = [−10, 80], pd = 0.93; Fig. 6b) but there was no evidence that preparation variability was affected by previous errors ($M_{diff} = 0$ ms, CI = [−60, 70], pd = 0.54; Fig. 6b).

Next, we explored the persistence of post-error effects over time. In particular, we examined the effect of inter-trial-interval on the post-error effect by comparing the results from Experiment 3 (0 ms ITI) with that of Experiment 4 (2000 ms ITI) in terms of the size of the post-error effect on response efficacy ($\beta$) (Fig. 7b). We observe that the post-error effect was greater in Experiment 3 ($\Delta_\beta = -0.0535$, CI = [−0.0802, −0.0314]) compared to Experiment 4 ($\Delta_\beta = -0.0141$, CI = [−0.0312, −0.00097]). This result suggests that although providing more time for stimulus processing and response preparation during a trial does

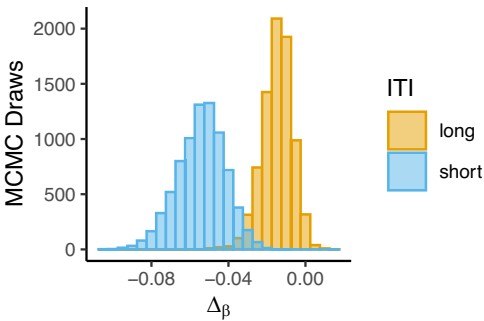

**Fig. 7 | ITI modulation of post-error effects.** Posterior distributions of post-error effects on response efficacy ($\beta$) for Experiment 3 (0 ms ITI) and Experiment 4 (2000 ms ITI). Experiment 3: $n = 47$ participants; Experiment 4: $n = 46$ participants.

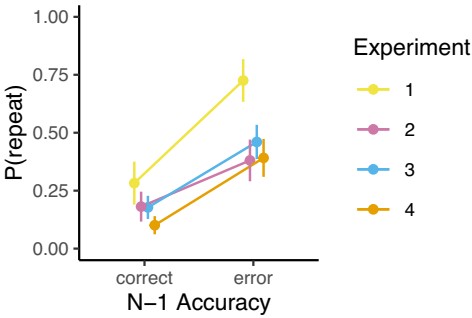

**Fig. 8 | Post-error perseveration.** Proportion of incorrect trials (PT > 1000 ms) in which the participant pressed the same key as on the previous trial, grouped by previous trial accuracy and experiment. Points are means, and error bars are 95% confidence intervals (±SEM * 1.96). Experiment 1: $n = 46$ participants; Experiment 2: $n = 37$ participants; Experiment 3: $n = 47$ participants; Experiment 4: $n = 46$ participants.

not eliminate post-error deficits, these post-error effects may dissipate during the time between trials.

### Exploring the nature of post-error processing deficits

Experiments 1–4 established that post-error deficits in performance are not due to the slowing ($\Delta\mu$) of cognitive processing underlying response preparation or an increase in trial-to-trial variability ($\Delta\sigma$) in the speed of cognitive processing. Instead, we observed more slips of action ($\Delta\beta$) at all time points where a correct response was likely to be prepared. What could be driving this static impairment in executing the correct response? One possibility is that people are simply biased away from repeating a response that just resulted in an error. We explored whether incorrect responses after an error were driven by a response bias—that is, a tendency to avoid/repeat the same key as in the previous trial. This analysis focused on incorrect trials for which the preparation time was greater than 1000 ms because this was the locus of the post-error effect. We fit hierarchical Bernoulli regression models to these data with repetition as the outcome and previous error as the covariate. Across all four datasets, we found that people were *more* likely to repeat the previous keypress after an error compared to after a correct trial (Fig. 8). We found strong evidence for increased perseveration after an error in Experiment 1 ($b = 1.03$, CI = [0.03, 1.96]), Experiment 2 ($b = 0.87$, CI = [0.19, 1.53], Experiment 3 ($b = 1.33$, CI = [0.76, 1.90]), and Experiment 4 ($b = 1.72$, CI = [1.03, 2.43]). It is not the case that these perseverative errors were simply due to initial perseverative guesses, as post-error perseveration makes up the larger percentage of all errors at the longer PTs relative to the shorter PTs (see Supplementary Table 1). These results suggest that an

increase in perseverative action slips following an error could at least partly explain post-error deficits in performance.

Experiments 3 and 4 established that performance recovers somewhat when there is more time to recover from an error before the start of the next trial. However, our participants were still more prone to slips of action after an error when given a full two seconds between trials. When does performance fully recover? To address this question, we explored the persistence of post-error effects across trials. In particular, we examined the lingering effect of errors on the subsequent two trials. We were especially interested in whether post-error effects persisted when there was an intervening correct trial. We re-fit the response preparation model with $N-2$ error as the covariate, separately for $N-1$ correct and $N-1$ error trials. For trials preceded by a correct response ($N-1$), we found little evidence for $N-2$ error effects (Exp. 1: $\Delta\beta = -0.008$, CI = [−0.021, 0.001]; Exp. 2: $\Delta\beta = -0.002$, CI = [−0.013, 0.009]; Exp. 3: $\Delta\beta = -0.017$, CI = [−0.036, 0.0001]; Exp. 4: $\Delta\beta = -0.017$, CI = [−0.033, −0.004]). However, for trials preceded by an error ($N-1$), we found evidence for $N-2$ error effects on current trial response efficacy ($\beta$) in Experiment 1 ($\Delta\beta = -0.024$, CI = [−0.053, −0.005]) and Experiment 4 ($\Delta\beta = -0.055$, CI = [−0.095, −0.024]), but not in Experiment 2 ($\Delta\beta = -0.015$, CI = [−0.040, 0.004) or Experiment 3 ($\Delta\beta = -0.033$, CI = [−0.075, 0.002]). These results suggest that post-error effects are 'reset' if there is an intervening correct response, but they may persist (or even compound) if there is an intervening incorrect response.

### Timing errors

In addition to key-press errors, participants also made timing errors (100 ms too fast or too slow) on a substantial number of trials and received feedback to remind them to respond at the time of the imperative cue (Exp 1—36.9% of trials; Exp 2—30.6%; Exp 3—40.3%; Exp 4—35.8%). Although including these mistimed trials does not substantially alter the behavioral or modeling results (see Supplementary Fig. 3), it is necessary to rule out that these timing errors could be affecting the post-error effects we observed. Across all four experiments, we did not find evidence that timing errors led to an increase in key-press errors on the subsequent trial (see Supplementary Fig. 4). Additionally, key-press errors did not substantially affect our participants' propensity to respond on time on the subsequent trials (no consistent significant results following multiple comparisons correction; see Supplementary Fig. 5). These results suggest that timing errors and key-press errors are independent of one another and that timing errors are not significantly contributing to the post-error effects reported here.

## Discussion

In the present study, we examined post-error effects in a stimulus–response task in which the time available for response preparation was manipulated. Across four experiments, we found that participants' response accuracy was lower if they made an error in the previous trial. This deficit was observed even when participants were given ample time to prepare their responses. A model-based analysis revealed that the post-error effect on accuracy was driven by a decrease in the probability of expressing a response given that it was highly likely to be prepared ($\beta$). These erroneous slips of action were predominantly perseverative repeats of the button pressed on the previous trial. Our analyses ruled out that the post-error effect on accuracy is due to a slowing of the cognitive processing required to prepare a response ($\mu$) or variability in the latency of cognitive processing ($\sigma$). These results suggest that previous findings of post-error slowing are unlikely to be due to a decrease in the speed of cognitive processing underlying the preparation of responses to stimuli. Furthermore, our results indicate that delaying the initiation of a prepared response in free response time tasks could be an adaptive response to impaired efficacy of

cognitive processing after an error rather than a strategic shift along a single speed–accuracy curve.

Our conclusions rely on the view that response preparation and response initiation are independent motor control parameters[13–15]. People not only decide *what* response to make, they also decide *when* to make their response. Prior studies on post-error effects do not directly distinguish between selection and initiation, in their experimental designs or in their theoretical models. One issue is that these studies use free RT, which is a combination of the duration of time spent selecting a response and the duration of delay after selection before initiation. Typical models used to explain post-error effects, such as the drift-diffusion model, are models of how people decide *what* to do—i.e., response selection. However, recent work in motor control shows that people initiate responses (in a free RT task) long after they have prepared those same responses as revealed in a forced RT task[13]. Thus, free RTs could be used to disentangle selection and initiation only with a more complete model that also describes the processes underlying the decision about *when* to initiate a selected response. Absent such a model, an alternative approach is to control response initiation using a forced-RT task, as in the present study.

Many studies in the past have shown that people respond more slowly after an error[2,7,18]. A prominent account of these effects is that participants are more cautious after an error[7]. For example, evidence accumulation models have been used to argue that participants alter their decision thresholds to accumulate more evidence before deciding on a response[5]. In this view, responses might take longer to prepare after an error. In the present study, we found that participants responded *less* accurately after an error, even if they were given up to two seconds to prepare their response. Contrary to some previous accounts, we found no evidence that the cognitive processing required to prepare responses occurred more slowly after an error. That is, our estimate of the latency at which a response is prepared ($\mu$) was unaffected by errors. Our data provide evidence against the view that post-error effects on performance are due solely to an increase in the evidence required to select a particular response. From the perspective of evidence accumulation, our forced response approach removes the ability for our participants to select a decision threshold as they must respond within 100 ms of the 'go' signal. Their responses, therefore, reflect the balance of the evidence for each response at each PT. Despite fixing decision thresholds, we still observe maladaptive post-error effects in the form of an increase in the propensity for action slips. This suggests that post-error slowing and increased decision thresholds observed in prior work might be compensating for current impairments in processing rather than simply just a reaction to negative feedback. However, it remains possible that post-error slowing and the erroneous perseverative responses could be independent effects following errors.

Other researchers have similarly suggested that there are unavoidable negative consequences to errors and unexpected events[6,8,10,11]. Notebaert and colleagues[6] proposed that infrequent, surprising events might cause an 'orienting response' that distracts participants from the processing of a subsequent stimulus. A similar account, dubbed the 'adaptive orienting theory,' advanced the idea that errors trigger a transient global inhibitory response that affects both motor and cognitive function that then gives way to adaptive cognitive processing to address the source of the error[10,11]. One might assume that this inhibition would lead to a change in the latency of cognitive processing underlying responses. Although similar in providing evidence for a maladaptive response following errors, our results do not fully support either of these accounts. From this prior work, one would expect that when ample time is given to make a response, we should observe identical levels of accuracy following both correct and incorrect responses or perhaps even increased accuracy following errors. An orienting response and a transient inhibitory response should resolve relatively quickly, and performance

should recover. Although we did find some evidence that a longer inter-trial interval lessened the deleterious impact of errors on subsequent performance (Experiments 3 and 4), participants' accuracy did not improve when they were given up to two seconds to respond following stimulus presentation. Purcell and Kiani observed more slowed responses following errors at low stimulus strength in a motion discrimination task[8]. Using drift-diffusion modeling of response time distributions, they described this result as a combination of an increase in the response threshold and a decrease in sensory signal-to-noise ratio (SNR) following errors. This would make the preparation of response less accurate, but also either slower or more variable. However, here we did not observe any slowing in the estimated latency ($\mu$) of the cognitive processing underlying response preparation following an error. We also did not find evidence of any increase in variability in the time it takes to complete this cognitive processing ($\sigma$). Instead, regardless of the amount of time given to prepare, participants were less accurate and were more prone to perseverative slips of action. Although the stimulus remained on the screen until a response was made in our task, it is possible that errors cause an initial impairment in subsequent stimulus–response processing that cannot be corrected online.

It is important to note that the results presented here do not rule out the fact that, in addition to maladaptive mechanisms following an error, there are likely to be adaptive mechanisms as well. Indeed both behavioral and neuroscientific investigations of post-error effects in recent years have provided evidence that errors can lead to subsequent attentional enhancement of task-relevant stimuli[19–21]. These studies all provided some support for the adaptive orienting account whereby initial impairments in performance in the moments just following an error give way to adaptive increases in attention and control processes several hundred milliseconds later. However, as we state in the paragraph above, it does not seem that these adaptive processes eliminate the effect of all maladaptive processing following errors. It is unclear if the post-error deficits in processing we observe here are independent of orienting responses reported previously. We do not have evidence from our results for a categorically different class of errors at short intertrial intervals as all post-error effects reside in the same parameter of our model ($\beta$), but it is possible that multiple separate maladaptive processes all lead to a higher propensity for action slips. We should also note that these prior studies all used conflict tasks that require participants to respond based on a task-relevant stimulus while resisting distraction from simultaneously presented task-irrelevant stimuli associated with a different response. It is possible that the nature of both maladaptive and adaptive mechanisms following an error might differ in tasks that don't require selective attention and processing in the face of distraction.

What leads to the slips of action we observe following errors? The model we use here to fit our data is agnostic as to the source of these errors and simply embodies the assumption that errors sometimes occur even when ample time is given for all the relevant cognitive processing required to make a correct response. Although we cannot rule it out entirely, given our data, we do not believe that these action slips are driven by issues with perceptual processing. At the long PTs, where we observe consistent reductions in accuracy error trials, the stimulus has been present on the screen for close to two seconds. Unless our participants were actively closing their eyes or looking away from the computer screen, this amount of time should have been plenty for perceptual processing to occur. It could be the case that the correct response is simply not prepared on a small subset of trials, or perhaps there is a memory error in retrieving the correct stimulus–response mapping. Action slips seem most likely to be due to a failure in action selection when perceptual information must be translated into a motor response.

The experiments used in the present study differ in one critical way from virtually all previous experiments examining post-error

effects: the time at which participants initiate their responses was controlled. This so-called interrogation method disables participants from strategically delaying the initiation of their responses, including after an error[22]. We found that, under this constraint, participants were less accurate after an error. A response preparation model explained this effect in terms of a decrease in the efficacy of the cognitive processing underlying prepared responses. In the model, "efficacy" is the parameter that controls the probability that a participant will express a response after it has been prepared. Although there are several ways to interpret this parameter, a natural psychological interpretation of efficacy is participants' confidence in their selected response. It may be that if people are led astray by decisions in the past, they become less confident about their decisions in the future. This reduced confidence would make them less likely to act on their decisions and potentially more prone to make random responses. Note that this is distinct from requiring more processing time to prepare a response. As noted above, we found little evidence to support post-error slowing of processing demands ($\mu$).

It is well-known that free RTs are not normally distributed and instead tend to follow distributions incorporating some skewness, such as the ex-Gaussian distribution[23]. Although allowing us to easily interpret modeling outcomes, one potential limitation of our study is that we assume that the amount of time it takes to complete the cognitive processing necessary to select a response can be approximated well by a normal distribution. However, we do not believe this limitation greatly affects the conclusions drawn here. The fit of our model to the data is quite good, perhaps specifically because the time deadlines do not allow for the long tail sometimes observed in RT distributions. Additionally, if it were the case that errors caused an increase in the skewness of the latency of cognitive processing, this would not come out as a reduction in the efficacy of cognitive processing ($\beta$), as we observed here. Instead, we would have observed a shift in the mean ($\mu$) and/or standard deviation ($\sigma$) of our estimates of the response preparation distributions to try to account for a longer tail. We found no evidence of this across our four experiments.

Our interpretation of the results relies on the assumption that cognitive processing in free RT tasks is similar to that seen in the forced response paradigm used here. We believe this assumption is reasonable. In the context of conflict tasks (e.g., the Simon task), similar effects on accuracy have been observed in a forced response paradigm as is observed in response time in free RT tasks[24]. However, it is possible that forcing participants to respond at a predetermined time changes the nature of the task and the cognitive processes underlying responses. For example, a forced response paradigm might lead to more task engagement and leave individuals less prone to inattentiveness. Participants in our paradigm must also monitor the response time cues in order to respond at the appropriate time. It is possible that this makes the task somewhat more difficult than free RT tasks that have investigated post-error effects. On the other hand, overall error rates on our task (~75%) are similar to other studies investigating the effects of errors on cognitive processing (e.g., [8,20]). Nevertheless, there is no particular reason to privilege data from free RT tasks when attempting to understand post-error effects on performance. The results presented here are not readily explained by prevailing theories of post-error effects that have been developed from free RT tasks. The forced response paradigm provides a window into post-error impairments in cognitive processing that have been difficult to examine with standard techniques.

One standard control analysis used in free RT investigations of post-error effects is to examine RT on trials preceding errors[1]. The upshot of this type of analysis is to ensure that any slowing of RT that is observed following an error is due to the commission of an error and not simply a result of lapses in attention where clusters of trials have both slower RTs and reduced accuracy. That is, if RT on correct trials preceding errors is also slower than the average RT of correct trials across the entire experiment, any post-error slowing that is observed is less likely to be due to the error itself. Most studies, however, show that RT on trials preceding errors is faster than the average correct response[25]. One shortcoming of the forced-response approach we adopt here is that we cannot perform an analog of this control analysis that independently examines trials pre- and post-error. As we only have a single dependent variable of interest (accuracy), once we uncover that trials following an error ($N + 1$) are more likely to display reduced accuracy and that errors tend to follow one another, it is necessarily true that trials preceding an error ($N - 1$) are also more likely to display reduced accuracy. Thus, looking at accuracy alone makes it more difficult to rule out the notion that the results presented here are due to lapses in attentiveness that span multiple trials. However, if the reduced accuracy rate we observed following errors were due to multi-trial periods of inattentiveness, we would also expect to observe other deficits in performance. For example, more errors and variability in the timing of responses. We did not find such an effect. Key-press errors and timing errors appear to be independent in our data and we did not observe a higher likelihood of timing errors on trials adjacent to key-press errors. This gives us more confidence that the increased slips of action we observe following errors are a result of post-error processing, though we cannot completely rule out that there may be some other process underlying the serial dependence we report here.

In sum, we provide evidence against the view that the cognitive processing necessary to translate a stimulus into a response is slower after an error. Instead, our data suggest that decisions about what to do are less likely to be translated into the appropriate motor responses after an error and lead to more perseverative slips of action. These results suggest a change in the shape of the speed-accuracy tradeoff function following errors and cast doubt on the idea that post-error slowing reflects increased caution that moves an individual along the same speed-accuracy curve.

## Methods
### Participants
Participants were recruited online using the online platform Prolific with the following inclusion criteria: US or Canadian nationality, fluent English speaker, and approval rate >95%. In Experiment 1 there were 46 participants (24 female) with a mean age of 33 years old. In Experiment 2, there were 37 participants (33 female) with a mean age of 27 years old. In experiment 3 there were 47 participants (24 female, 1 declined to answer) with a mean age of 29 years old. In Experiment 4, there were 46 participants (22 female, 3 declined to answer) with a mean age of 32 years old. No statistical method was used to predetermine sample size, though pilot work with smaller sample sizes was sufficient to obtain reliable results. All research protocols were approved by the Health Sciences and Behavioral Sciences Institutional Review Board at the University of Michigan. All participants gave written informed consent. Participants were paid $10/h for their participation.

### Experiments
First, participants trained for 60 trials in a stimulus–response task. For this task, participants were instructed to press 'f', 'g', 'h', or 'j' depending on the identity of the stimulus (see below). At the start of each trial ($t = 0$ ms), a stimulus was presented, and the identity of the stimulus was randomly sampled from a uniform categorical distribution over four unique stimuli (colors or symbols). During each trial, the four possible stimulus–response mappings were presented at the bottom of the screen to help participants learn these mappings. After each trial, participants were given feedback for 500 ms about the accuracy of their responses. Response times were unconstrained during this phase of the experiment.

Next, participants trained for 20 trials in a fixed response timing task. For this task, participants were instructed to press a key exactly when two empty white rectangles were filled completely in color,

exactly two seconds after the trial began. At the beginning of each trial, two empty rectangles (PsychoPy height unit: .35*0.03% of the screen) were shown above and below where the stimuli had appeared in the previous training block. Every 500 ms, the rectangle was filled in by an additional 25%. After 2100 ms, all stimuli were removed from the display. The purpose of this cuing was to guide participants to respond at the same time on every trial. Participants were encouraged beforehand to alternate between 'f', 'g', 'h', and 'j' to practice timing with all keys. After each trial, participants were given feedback for 500 ms about whether they responded too quickly (RT < 1900 ms), too slowly (RT > 2100 ms), or with perfect timing.

Finally, we turned to the main experimental task, illustrated in Fig. 1. Participants performed 10 blocks of 40 trials in the stimulus–response task with a fixed response timing and a stimulus presentation time that was parametrically varied. As in the first phase of training, participants were instructed to press one of four keys depending on which one of the four stimuli was shown. The onset and identity of the stimulus varied randomly, as in the first phase of training. However, there was no prompt showing the S–R mappings during the trial. As in the second phase of training, participants were instructed to respond only when the rectangle timing cue was filled. This approach allows us to measure the accuracy of responses when the exact amount of time allowed for stimulus processing and response preparation is known. After each trial, participants were given feedback, but the exact specifications of this feedback varied across experiments.

In Experiment (Exp.) 1, the target stimuli were letters in the Armenian alphabet (Ո, Ճ, Ջ, Ձ; PsychoPy height unit: 0.2% of screen) and participants were given feedback for 1000 ms about whether they were too slow, too fast, or had perfect timing, followed by a 1000 ms inter-trial-interval (ITI). Exp. 2 was an exact replication of Exp. 1.

In Experiments 3 and 4, the target stimuli were color-filled circles (orange, blue, red, or purple; PsychoPy size: 0.2 × 0.2). Here the response keys were 'd', 'f', 'j', and 'k'. Participants were given 400 ms feedback if and only if they responded too quickly or too slowly; no feedback if their timing was perfect. Feedback was followed by an ITI of 0 ms in Exp. 3 and an ITI of 2000ms in Exp. 4.

All experiments were built using PsychoPy/JS and run online using Pavlovia.

## Pre-processing

Our analyses focused exclusively on behavior in the test phase of the experiment. We excluded trials in which participants responded too quickly (RT < 1900 ms) or slowly (RT > 2100 ms) because we were interested in how people behaved when their response times were fixed to the imperative cue and to eliminate any post-error slowing of response emission itself (see Supplementary Fig. 2). This filtering step removed 36.9% of trials Experiment 1, 30.6% of trials for Experiment 2, 40.3% of trials for Experiment 3, and 35.8% of trials for Experiment 4. Including these mistimed trials in the analyses does not substantially alter the behavioral or modeling results but does make the interpretation of these results more challenging, given that we are no longer interrogating a participant's cognitive state at a predetermined time. Since we were interested in the effects of immediately preceding errors, we also excluded trials for which there was no immediately preceding trial (i.e., the first trial of each block for each participant). Response accuracy ($y$) was set to 1 if the response was correct and 0 if the response was incorrect. Preparation time (PT) was defined as the duration between the stimulus onset and the response time. PT was rescaled range from zero to one by dividing by 2000 to facilitate Bayesian prior specifications. Each trial was labeled with the outcome of the previous trial. Previous error ($err_{n-1}$) was set to 0.5 if the previous response was incorrect and −0.5 if the previous response was correct so that the intercepts in models with post-error slopes can be interpreted as the average across levels of the condition.

## Analysis

We used a sliding window technique to visualize the mean of response accuracy as a function of PT. The sliding window was performed separately for each level of $err_{n-1}$. The width of the sliding window was 100 ms, the step size was 1 ms, and the window moved from 0 to 2000 ms.

We used Bernoulli regression models to assess the credibility of apparent effects in the smoothed conditional accuracy functions. The models had hierarchical slopes and intercepts (i.e., participant-level variables sampled from group-level distributions) and focused on specific contiguous intervals over PT (e.g., 0 ms < PT < 500 ms). We used the R-package brms[26] to specify the models and to approximate posterior distributions over unobserved variables, given the observed data. From these distributions, we report the median value, the 95% credible interval (CI), and the probability of direction (pd). Given the observed data, the effect of interest has a 95% probability of falling within the range of values specified by the credible interval. The probability of direction (ranging from 0.5 to 1.0) refers to the probability that an effect goes in a particular direction (i.e., negative or positive).

We used a response preparation model to explain the observed time courses of performance[15]. This model enables inferences about the speed with which responses were prepared ($\mu$), the variability in this preparation ($\sigma$), the probability that a prepared response will be expressed if it is prepared (efficacy, $\beta$), as well as effects of covariates ($\Delta$) on these variables. This model also includes a parameter that governs chance level accuracy when no response has been prepared ($\alpha$). We specified the model using Stan[27]. In this model, the unobserved variables mentioned above were computed as linear functions of $err_{n-1}$, and the effects of $err_{n-1}$ were captured by a set of delta parameters. All intercepts and slopes in the model ($\mu_0$, $\Delta_\mu$, $\sigma_0$, $\Delta_\sigma$, $\beta_0$, $\Delta_\beta$) were hierarchical (i.e., participant-level variables sampled from group-level distributions), and we assigned weakly informative priors to the group-level variables. The model specification is reported below, where $y$ is the observed response accuracy, $p$ is the probability of correct response, $f$ is the response preparation function which computes $p$ given the observed preparation time $t$ and the unobserved variables $\theta = \{\mu, \sigma, \beta, \alpha\}$, and $\theta_{jk}$ is the unobserved variable $\theta$ for participant $j$ and condition $k$ (i.e., post-error level).

$$y \sim \text{Bernoulli}(p) \tag{1}$$

$$p = f\left(t, \mu_{jk}, \sigma_{jk}, \beta_{jk}, \alpha_j\right) \tag{2}$$

$$f \leftarrow \varphi \cdot \psi \tag{3}$$

$$\varphi = \left[1 - \text{Normal}_{\text{cdf}}\left(t, \mu_{jk}, \sigma_{jk}\right), \text{Normal}_{\text{cdf}}\left(t, \mu_{jk}, \sigma_{jk}\right)\right] \tag{4}$$

$$\psi = \begin{bmatrix} \alpha_j \\ \beta_{jk} \end{bmatrix} \tag{5}$$

As mentioned above, unobserved variables (except for α) were computed as linear functions of $err_{n-1}$. Below is the specification of these variables, where $\theta_j^0$ is the intercept for the variable $\theta$ and participant $j$, $x$ is the value of $err_{n-1}$, $\Delta_j^\theta$ is the effect of $x$ on $\theta$ for participant $j$. Inverse-logit link functions were used to constrain the values of the variables between 0 and 1 (seconds for $\mu$ and $\sigma$, probability for $\alpha$ and $\beta$).

$$\mu_{jk} = \text{logit}^{-1}\left(\mu_j^0 + \Delta_j^\mu * x\right) \tag{6}$$

$$\sigma_{jk} = \text{logit}^{-1}\left(\sigma_j^0 + \Delta_j^\sigma * x\right) \tag{7}$$

$$\beta_{jk} = \text{logit}^{-1}\left(\beta_j^0 + \Delta_j^\beta * x\right) \tag{8}$$

$$\alpha_j = \text{logit}^{-1}\left(\alpha_j^0\right) \tag{9}$$

The slopes and intercepts of the variables above were defined hierarchically. Below is the specification of these hierarchical variables, where $\theta_{\text{loc}}^0$ is the group-level mean of the intercepts for a parameter $\theta$, $\theta_{\text{scale}}^0$ is the group-level standard deviation of the intercepts, $\Delta_{\text{loc}}^\theta$ is the group-level mean of the err$_{n-1}$ effects on $\theta$, and $\Delta_{\text{scale}}^\theta$ is the group-level standard deviation of the err$_{n-1}$ effects. Note that in the stan code, we used a non-centered parameterization, despite presenting the centered parameterization below for ease of comprehension.

$$\mu_j^0 \sim \text{Normal}\left(\mu_{\text{loc}}^0, \mu_{\text{scale}}^0\right) \tag{10}$$

$$\Delta_j^\mu \sim \text{Normal}\left(\Delta_{\text{loc}}^\mu, \Delta_{\text{scale}}^\mu\right) \tag{11}$$

$$\sigma_j^0 \sim \text{Normal}\left(\sigma_{\text{loc}}^0, \sigma_{\text{scale}}^0\right) \tag{12}$$

$$\Delta_j^\sigma \sim \text{Normal}\left(\Delta_{\text{loc}}^\sigma, \Delta_{\text{scale}}^\sigma\right) \tag{13}$$

$$\beta_j^0 \sim \text{Normal}\left(\beta_{\text{loc}}^0, \beta_{\text{scale}}^0\right) \tag{14}$$

$$\Delta_j^\beta \sim \text{Normal}\left(\Delta_{\text{loc}}^\beta, \Delta_{\text{scale}}^\beta\right) \tag{15}$$

$$\alpha_j^0 \sim \text{Normal}\left(\alpha_{\text{loc}}^0, \alpha_{\text{scale}}^0\right) \tag{16}$$

Finally, we assigned weakly informative priors to group-level location and scale variables. Scale variables were constrained to be positive and assigned Normal (0, 0.5) priors. Location variables were left unconstrained. The prior was Normal (−0.5, 0.5) for the location of the μ intercept, Normal (−2, 0.5) for the σ intercept, Normal (2, 0.5) for the β intercept and Normal (−1, 0.5) for the α intercept. Delta location variables were assigned Normal (0, 0.5) priors. Note that these normal distributions are akin to beta distributions after being transformed by the inverse logit function. Though these variables are all assumed to be independent, our approach here allows for the possibility that the data are best fit when variables are correlated with one another (e.g., slower μ coincides with a larger σ).

To assess model fit, we first visualized posterior predicted behavior to compare to participants' data qualitatively (see Supplementary Fig. 1). Given that the results across all four experiments implicated changes in the β parameter following errors, we next performed a formal model comparison between the full model and a simpler model that only allows μ and σ to vary (i.e., including $\Delta_\mu$ and $\Delta_\sigma$, but omitting $\Delta_\beta$). Models were compared based on their out-of-sample predictive fit via expected log pointwise predicted density (ELPD) with a relative weight assigned to each model. This weight can be interpreted as the probability of each model given the data. Across all four experiments, the full model was assigned a higher weight than the simpler model with a static β (Exp. 1−relative weight: 0.900, 10 times more likely than the simpler model; Exp. 2−0.663, 2 times more likely; Exp. 3−0.928, 13 times more likely; Exp. 4−0.794, 4 times more likely).

No sex or gender analysis was carried out as there is no prior evidence that post-error effects differ by sex or gender and we had no hypotheses as to the effect of these variables.

## Reporting summary

Further information on research design is available in the Nature Portfolio Reporting Summary linked to this article.

## Data availability

The datasets generated and analyzed by the studies presented here are available in an Open Science Framework repository: https://doi.org/10.17605/OSF.IO/U7SZR[28].

## Code availability

The computer code used for the analyses and figures can be found in an Open Science Framework repository: https://doi.org/10.17605/OSF.IO/U7SZR[28].

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

## Acknowledgements

We thank John Jonides for his advice and feedback on this project. Salary support for Han Zhang was provided by grants from the National Institutes of Health (R21MH129909) and the National Science Foundation (Grant No. 2238151).

## Author contributions

T.A. designed, and performed experiments, analyzed data, and wrote/revised the paper. H.Z. designed and performed experiments, analyzed data and revised the paper. T.L. designed experiments, evaluated data, wrote/revised the paper, and supervised the project.

## Competing interests

The authors declare no competing interests.
