## [Peer Review File · Nature Communications]

People are more error-prone after committing an errorReviewers' comments:

Reviewer #1 (Remarks to the Author):

This paper considers the question of how is affected following an error. Previous work suggests that people slow their behavior after an error and this is thought to reflect heightened caution. The authors make a good case that these findings, based on free-reaction-time approaches may be misleading and instead they use a forced-response paradigm to measure in detail the time required to select and prepare the correct response to the stimulus. They find that, in trials that follow an error, the process of response selection is not slower or more variable following an error. Participants are, however, more likely to produce an incorrect response, independent of the amount of time they are allowed to prepare their movement. Examining these errors more closely, the authors find that the errors reflect a pattern of perseveration of responses from previous trials.

Understanding how errors affect our behavior has important implications in applied domains as well as in understanding and controlling for the impact of errors in experiments. This paper provides an innovative take on this topic and, I believe it, generates important new insights. Overall, I am positive about the contribution of this paper, though I have few concerns.

One concern I have is with the interpretation of the reduced rate of correct responding. This effect is encapsulated in the efficacy parameter β , and is where the authors find the major effect of errors on subsequent behavior. In most parts of the manuscript, the authors suggest that the action was correctly prepared, but that a "slip of action" at the time of response initiation led to a different response being generated. For example, line 75: " $1-\beta$ is the probability that an action slip occurs even if the correct response has been prepared". See also lines 90-92 and many other parts of the paper. While this could be the case, the errors might also reflect trials in which responses are mis-prepared, or in which preparation in response to the stimulus fails to occur at all with participants instead just holding on to their initial guess.

What was the probability of perseveration for errors on trials with very short preparation times? My guess is that you would find a similar effect here as for long preparation times, implying that participants tend to make perseverative guesses, and are more likely to lapse on re-preparing a different response post-error - perhaps because they are still ruminating on the error. This relates to the issue of whether the perseverative errors are due to slips-of-action that creep in at the last minute, or are due to lapses in stimulus processing and/or re-preparation. These two could potentially be dissociated if the distribution of guesses at low RT are different from the distribution of failures at high RT on post-error trials.

Another concern is with the fact that the reported results of the analysis don't entirely seem to align with the plotted accuracy curves. In Experiment 3 (and to some extent, Experiment 2), it appears from the response accuracy curves that performance is slower following an error - the curve is clearly shifted rightwards. I have a hard time squaring this observation with the result from the analysis which concludes that there is no hint of an effect of errors on μ (the confidence interval for $\Delta\mu$ is symmetric around zero). This somewhat undermines confidence in the overall statistical analysis. Can the authors offer some explanation as to how this apparent effect in the figures vanishes in the analysis?

One possible issue with the model is that μ and σ are not really independent, but are likely to be correlated, e.g. if the accuracy curves start increasing from around the same time point but increase at different rates, this would be described by a change in both μ and σ . My understanding of the analysis is that these are treated as independent. How might this affect the outcome of the analysis?

Since the results are presented before the methods, it would be useful to include a brief overview

within the results of how the hypotheses were actually tested (i.e. by fitting a hierarchical Bayesian model to population data). Exactly how to interpret the results is not clear, especially without reading the methods first. For instance, the quantity M_{diff} for the mean and variance parameters is in fairly arbitrary coordinates that are pretty meaningless without knowing that the time was normalized to a [0,1] range. Even then, reporting it in this way obfuscates the actual result. Couldn't this just be converted to ms so that it's something the reader can meaningfully interpret? It's also unclear exactly what "pd" means.

"Preparation Noise" is a slightly unclear terminology - the preparation itself is not corrupted in any way, just delayed. There doesn't seem to be any rationale why this effect should be interpreted as being due to noise. In fact arguably, the "efficacy" parameter could be regarded as preparation "noise" as that is there the preparation is actually corrupted (in one interpretation at least). The authors might consider describing this as "variability" in the time required for preparation, rather than "noise".

line 25 - "Post-error slowing sometimes coincides with increased response accuracy"

line 32 - "Accuracy is quite often stable or even reduced after an error"

These two statements from the introduction are rather vague and contradictory.

line 42 - This sentence was difficult to parse

line 394 - typo

line 454 - "will BE expressed"

Reviewer #2 (Remarks to the Author):

The authors propose an experimental "interrogation" design where subjects must respond at a given time (in contrast to more standard free RT paradigms). Using this design and corresponding model, they find that an error on the previous trial determines the probability of a slip (ie., an error despite a correct response preparation), but not other parameters of the model.

I find this an interesting paper, but I have several comments on the paradigm, the model, and how the authors interpret their results (very roughly in decreasing order of importance).

The authors only report model parameters, but can we also get some estimate of model fit (either absolute or relative to other models)? Also, can you show some simulations of the model (based on estimated parameters)? Except if I missed it, they are not presented in the paper.

The fact that the effect resides in beta, can it be an artifact of the fact that at early PTs, there is simply no effect possible? Indeed, one is near floor level performance then. More generally, it would be nice to show via simulations (or perhaps earlier data) that effects can in principle reside in any of the parameters of the model.

I find the distinction that is made in the model between "cognitive processing" (a normally distributed variable with a mean and a variance) and the efficacy of cognitive processing (beta), quite arbitrary. What warrants this distinction? Are there different brain areas assigned to these two aspects, or is it only a mathematical convenience? Is there evidence that this type of model fits better than the more established type of sequential sampling (e.g., diffusion) model (see previous point). Of course, not everybody has to use diffusion models, but the fact that this type of model cannot even produce a positively skewed RT distribution, seems a big disadvantage to me. The rather subtle conceptual (and non-empirical) points that the authors make in this regard about deciding when to emit a response, are not really convincing to me. Relatedly, is the assumption that when cognitive processing is "ready", a correct response is always given (but that this correct response can accidentally slip)?

Line 186: If ... then these effects might disappear with a sufficiently long ITI. And the current theory does not "predict" that these effects might disappear? I do find the experiment informative, but I don't see how it would disentangle the current theory from the orienting theory.

Also related to this point, the novel task that the authors present is actually a dual-task, because subjects must pay attention to the RT cue while waiting for / processing a stimulus. In this sense, the gain of not having the "when to respond" process (as the standard design presumably has), also comes with a cost of more difficult / less natural task.

One other disadvantage (or feature, depending on how looks at it), is that error probability is much higher than in standard designs. Thus, an error is also not so bad, or so unexpected as it typically is. It would be great if the authors can comment on this (or even better, present an analysis that addresses it).

It would also be interesting to see the effect of the type of previous error. For example, making an error if one has 0 msec processing time available (i.e., on the previous trial), seems cognitively very different than if one had 1000 msec available (again, on the previous trial).

The authors might also consider exchanging error on trial $n - 1$ with error on trial $n + 1$. This is a standard type of procedure in the post-error slowing literature; indeed; if the trial $n - 1$ effect remains with $n + 1$ trials, it cannot be a causal effect (and is more likely due to a slow drift across trials). Of course, this is not a standard type of design (with much larger range of error rates than usual), but could still be of interest.

Preprocessing: I may have missed it, but what % of trials is removed by excluding responses outside the response window? Were subjects informed that they were too early/late in such case?

Last sentence of abstract: PES may be A instead of B; A and B seem simply different formulations of the same concept to me. This makes me wonder what exactly has been shown by the paper. On line 92, a similar claim is made again. To clarify, the point of shifting along the speed accuracy curve is exactly to adapt (become slower but more accurate) in response to impairments in cognitive processing (here, after an error).

Line 42: "that are analyzed separately": Usually, a diffusion model (or similar evidence accumulation model is fit)

Line 71: Shouldn't alpha also be described as a parameter? Or is that fixed?

Reviewer #3 (Remarks to the Author):

The authors of this manuscript present the results of a model-based analysis of four behavioral online experiments addressing adaptive or maladaptive changes in information processing on trials following the commission of an error. They make use of a forced-response paradigm in which they manipulate the duration of stimulus processing prior to the forced response and the inter-trial interval (ITI). A response preparation model was built and free parameters indicating the speed and variance of response preparation and the efficacy of response execution were fit to the behavioral data. While response preparation was not significantly modulated after errors, efficacy of response execution was impaired and more perseverative slips were committed after errors. The authors suggest that these findings are not fully compatible with current theories of post-error adjustments and, in particular, post-error slowing.

The study addresses a research topic that has been debated for a long time and seems to add an

interesting result to the literature. However, in my opinion the conceptual advancement provided by this manuscript is rather limited. The results do not seem to lead to a novel theoretical account that can explain the quite variable findings that have been reported previously based on a breadth of tasks. In other words, it remains unclear to me how the findings and conclusions can be extended to other tasks; they seem to be quite specific to the task studied here. Therefore, I believe that the results will not be of sufficient interest to the broad readership of this journal and might be better suited for more specialized journals. It also appears that the authors need to clarify the concepts and terms they use in the manuscript. I, furthermore, have a number of questions and concerns that are listed below:

1. Abstract: I think it is not fair to state the the dominant account of post-error slowing is to assume more cautious responses after errors. This view has been challenged multiple times and disentangling adaptive and maladaptive adjustments after errors has been the topic of many theoretical and empirical papers over the last 15 years or so. Some of these are, while being quite relevant, not mentioned in the manuscript, e.g., Wessel (2018) An adaptive orienting theory of error processing, *Psychophysiology*; Fischer et al. (2018) Cortical beta power reflects decision dynamics and uncovers multiple facets of post-error adaptation, *Nature Communications*; Steinhauser et al. (2019) Rapid adaptive adjustments of selective attention following errors revealed by the time course of steady-state visual evoked potentials, *Neuroimage*. Taking together the current literature suggests at least to me that adaptive as well as maladaptive mechanisms kick in after an error and the relative timing and impact of these mechanisms depends on the type of task and contextual factors. Also the definition of "adaptive" or "maladaptive" heavily depends on the context and the goals of the acting person.

2. The task is interesting and certainly adds a new perspective to the field. However, I am not sure whether it can be interpreted in such straightforward fashion as the authors suggest such that the "what" and "when" decisions could be unequivocally disentangled. It appears that participants have to do two tasks in parallel: (1) monitoring the square and pressing at the right time and (2) selecting the correct response. Thus, timing errors as well as action selection errors can occur and the analyses (as well as the model) do not address any potential interactions between both tasks. Other aspects such as response urgency (cf. work by Paul Cisek) is not taken into account. If the stimulus appears only briefly before the time point when the response should be executed, urgency (and response vigor) can be expected to very high. Again, multiple accounts for urgency have been suggested, e.g. an urgency signal, increases of gain in evidence accumulation and/or collapsing decision boundaries. Also, is it valid to assume that participants guess randomly, "if stimulus-based response preparation is not yet finished" (p.3, l.34) What is meant by complete here? What, if some evidence is already accumulated, then the response will be biased and not random. Do participants prioritize responding in time over selecting the correct button with sufficient confidence?

3. What could be the mechanisms of action slips if response selection is complete?

4. Important data is missing in the manuscript. The descriptive stats on error trial numbers and error rates, error types, timing errors etc. need to be reported. The trial numbers are important to assess the reliability and robustness of the analyses. How often did "too fast"/"too slow" feedback occur and how was their influence on subsequent trials taken into account? The methods section suggests that these trials were excluded. How many trials were excluded? Exclusion of trials also disrupts sequences of trials relevant for analysis (error-correct, correct-correct etc.). Was there a higher likelihood to make mistakes after timing feedback (too fast/slow)?

5. If participants prioritize response timing over accuracy, it does not seem to be surprising that no post-error slowing was found. They just try to be in time at any cost. This urgency effect could counteract slowing. Post-error slowing has been suggested to result from motor inhibition. In connectionist models (Botvinick et al, 2001) as well as sequential sampling models (e.g., drift diffusion models) post-error slowing is implemented via a shift of the decision boundary. Purcell & Kiani (2016) as well as Fischer et al. (2018) provided empirical evidence supporting this account. It is not clear why a shift in the motor threshold would necessarily prolong "cognitive processing" or "preparation speed".

And perhaps the threshold is not shifted in this task or any threshold shifts are compensated by a gain increase/an urgency signal.

It is indeed intriguing that error likelihood and perseveration is increased on post-error trials, but the authors do not seem to provide a theoretical account for that finding. What could lead to the effect that the previous response representation in the motor system is "overwritten" less efficiently after errors?

6. p. 11, l. 22: "Instead, post-error slowing effects might be better explained by adaptive delays in response initiation to compensate for impairments stimulus-response processing that lead to more frequent action slips" - this is purely speculative and not supported by the data. If this speculation were true, what mechanism would trigger that additional adaptive delay? And how could it compensate for more frequent action slips?

We thank the reviewers for their insightful comments. We believe the manuscript has improved substantially in light of the revisions in response to some of the issues raised in the review process. See below for a point-by-point response to reviewer comments (responses are in blue and italicized).

Reviewer #1 (Remarks to the Author):

This paper considers the question of how is affected following an error. Previous work suggests that people slow their behavior after an error and this is thought to reflect heightened caution. The authors make a good case that these findings, based on free-reaction-time approaches may be misleading and instead they use a forced-response paradigm to measure in detail the time required to select and prepare the correct response to the stimulus. They find that, in trials that follow an error, the process of response selection is not slower or more variable following an error. Participants are, however, more likely to produce an incorrect response, independent of the amount of time they are allowed to prepare their movement. Examining these errors more closely, the authors find that the errors reflect a pattern of perseveration of responses from previous trials.

Understanding how errors affect our behavior has important implications in applied domains as well as in understanding and controlling for the impact of errors in experiments. This paper provides an innovative take on this topic and, I believe it, generates important new insights. Overall, I am positive about the contribution of this paper, though I have few concerns.

One concern I have is with the interpretation of the reduced rate of correct responding. This effect is encapsulated in the efficacy parameter beta, and is where the authors find the major effect of errors on subsequent behavior. In most parts of the manuscript, the authors suggest that the action was correctly prepared, but that a "slip of action" at the time of response initiation led to a different response being generated. For example, line 75: "1-beta is the probability that an action slip occurs even if the correct response has been prepared". See also lines 90-92 and many other parts of the paper. While this could be the case, the errors might also reflect trials in which responses are mis-prepared, or in which preparation in response to the stimulus fails to occur at all with participants instead just holding on to their initial guess.

*We regret the lack of precision in our writing when discussing how the model assumes errors occur vs. what psychological processing could lead to these errors. The model is quite good at estimating the latency of response preparation from the data (as described by a normal distribution). In the model, the beta parameter is simply just the accuracy rate given that it is **very likely** that a response should have been prepared given the amount of processing time (PT) given (as estimated by the rest of the participant's data). Of course, the reviewer is correct that it is possible that it is possible that errors that occur when ample time has been given for response preparation might reflect mis-prepared responses or a failure of response preparation altogether on a small subset of trials. We intended the term "action slip" to subsume these possibilities, with the caveat the error is almost assuredly due to a failure in action selection when translating the stimulus to a response (rather than a problem with perceptual processing). The stimulus stays on the screen up until the response needs to be made. So at the longer PTs, the participant has seen the stimulus for over 1 second continuously (unless they were looking*

away from the screen). We have changed how we introduce the term “action slip” and have added to our discussion to go through the different possibilities for what might lead to these types of errors.

What was the probability of perseveration for errors on trials with very short preparation times? My guess is that you would find a similar effect here as for long preparation times, implying that participants tend to make perseverative guesses, and are more likely to lapse on re-preparing a different response post-error - perhaps because they are still ruminating on the error. This relates to the issue of whether the perseverative errors are due to slips-of-action that creep in at the last minute or are due to lapses in stimulus processing and/or re-preparation. These two could potentially be dissociated if the distribution of guesses at low RT are different from the distribution of failures at high RT on post-error trials.

Below is a table showing the percentage of errors in each processing time bin that were perseverative errors (i.e. repeats of the same button pressed on the previous trial). Although the pattern isn't entirely consistent across the four experiments, it seems that if anything post-error perseverative errors are MORE likely at the longer PTs. Particularly striking is the fact that it seems that guess responses at low PTs are more likely to be a perseverative response and this effect dissipates over time after correct responses. We do see some hint (in the first 3 experiments at least) that participants may have an initial ‘perseverative guess’ as the reviewer suggests, but it seems unlikely that this is driving the results given that the bulk of the difference between post-error trials and post-correct trials occurs at the later PTs when there has been ample time for stimulus processing.

% of Perseverative Errors (out of all errors)					
Processing Time (in ms)	Previous trial	Experiment 1	Experiment 2	Experiment 3	Experiment 4
0-500	Incorrect	33.96	31.05	39.38	22.02
501-1000	Incorrect	37.68	47.93	40.77	41.67
1001-1500	Incorrect	36.07	72.41	41.41	45.21
1501-2000	Incorrect	40.38	72.73	51.95	32.31
0-500	Correct	37.41	36.67	37.30	26.00
501-1000	Correct	28.26	30.16	20.36	17.34
1001-1500	Correct	15.73	30.00	20.15	10.42
1501-2000	Correct	22.45	25.00	14.29	9.52

As we note above, we find it unlikely that the errors governed by the beta parameter are due to lapses in stimulus processing. At the long PTs, the stimulus is continuously on the screen for up to 2 full seconds. We believe that we can be fairly certain that the visual processing has occurred and that the issue is translating this visual information into a response. The fact that the proportion of perseverative errors is large at these longer PTs seems to suggest that the post-error perseveration is not due to holding onto a perseverative guess, but rather must involve either a slip of action post-perceptual processing or perhaps a failure in translating the stimulus to the appropriate response. We have added this table as a Supplementary Result and have added to the Discussion section regarding these issues.

Another concern is with the fact that the reported results of the analysis don't entirely seem to

align with the plotted accuracy curves. In Experiment 3 (and to some extent, Experiment 2), it appears from the response accuracy curves that performance is slower following an error - the curve is clearly shifted rightwards. I have a hard time squaring this observation with the result from the analysis which concludes that there is no hint of an effect of errors on μ (the confidence interval for $\Delta\mu$ is symmetric around zero). This somewhat undermines confidence in the overall statistical analysis. Can the authors offer some explanation as to how this apparent effect in the figures vanishes in the analysis?

It can actually be very difficult to assess whether a rightward shift of the accuracy curve is due to slowing (a rightward shift) or increased errors/perseveration (a downward shift) by just inspecting the plots visually. This is why the modeling is so helpful! Plotted below is simulated data where either the μ (mean of the preparation time) is slowed down by 50 ms or there is simply more perseveration (i.e. decrease in β by 5%) compared to baseline. As you can see, both manipulations produce a seemingly rightward shift in the curve. The key difference is that when there is perseveration, but not slowing, the asymptote is at a lower level of accuracy. We find this lower asymptote across all 4 experiments and the data are very well described by a reduction in β alone. Model simulations of this sort appear in the bottom panels of Figure 2, but we now realize that it may have been unclear that these plots were simulated from varying the parameters systematically rather than just schematic illustrations. We have called attention to this fact when we introduce the figure in the text and in the figure caption.

One possible issue with the model is that μ and σ are not really independent, but are likely to be correlated, e.g. if the accuracy curves start increasing from around the same time

point but increase at different rates, this would be described by a change in both μ and σ . My understanding of the analysis is that these are treated as independent. How might this affect the outcome of the analysis?

Our analysis is actually agnostic to whether μ and σ are independent. The reviewer is certainly correct in that we do not impose any correlation between the two. However, if they both move in the same direction after errors, our model would be equally likely to pick up on it compared to a situation where they are anticorrelated or independent. Our model fitting does not sequentially examine each variable in isolation. It would be a bit more accurate to say that the model allows them to be independent or correlated if that is what provides the best fit to the data. We have amended the Methods section to make this clearer.

Since the results are presented before the methods, it would be useful to include a brief overview within the results of how the hypotheses were actually tested (i.e. by fitting a hierarchical Bayesian model to population data). Exactly how to interpret the results is not clear, especially without reading the methods first. For instance, the quantity M_{diff} for the mean and variance parameters is in fairly arbitrary coordinates that are pretty meaningless without knowing that the time was normalized to a $[0, 1]$ range. Even then, reporting it in this way obfuscates the actual result. Couldn't this just be converted to ms so that it's something the reader can meaningfully interpret? It's also unclear exactly what "pd" means.

We regret that our writing was unclear on these points. The $[0, 1]$ range we use for time was simply in seconds and was not normalized. However we now realize that this was confusing given that β also has a $[0, 1]$ and we report things in ms in most places and in seconds for M_{diff} . We have converted the reported results back to ms to address this.

The acronym "pd" refers to "probability of direction", which is used in Bayesian modeling to convey how much of the posterior distribution is positive or negative. It is strongly correlated with a p -value from frequentist statistics and it is intended to give the reader a sense of the evidence for an effect in a particular direction. We have added language to address this both in the methods and the first instance we use the acronym pd.

"Preparation Noise" is a slightly unclear terminology - the preparation itself is not corrupted in any way, just delayed. There doesn't seem to be any rationale why this effect should be interpreted as being due to noise. In fact arguably, the "efficacy" parameter could be regarded as preparation "noise" as that is there the preparation is actually corrupted (in one interpretation at least). The authors might consider describing this as "variability" in the time required for preparation, rather than "noise".

We agree that variability is likely a more precise term here. We have replaced "noise" with "variability throughout the manuscript.

line 25 - "Post-error slowing sometimes coincides with increased response accuracy"

line 32 - "Accuracy is quite often stable or even reduced after an error"

These two statements from the introduction are rather vague and contradictory.

line 42 - This sentence was difficult to parse

line 394 - typo
line 454 - "will BE expressed"

These errors have all now been corrected in the manuscript

Reviewer #2 (Remarks to the Author):

The authors propose an experimental “interrogation” design where subjects must respond at a given time (in contrast to more standard free RT paradigms). Using this design and corresponding model, they find that an error on the previous trial determines the probability of a slip (ie., an error despite a correct response preparation), but not other parameters of the model.

I find this an interesting paper, but I have several comments on the paradigm, the model, and how the authors interpret their results (very roughly in decreasing order of importance).

The authors only report model parameters, but can we also get some estimate of model fit (either absolute or relative to other models)? Also, can you show some simulations of the model (based on estimated parameters)? Except if I missed it, they are not presented in the paper.

We regret this omission and now include the posterior predicted accuracy of our model and figures showing how well our model with the estimated parameters fits our participants' data (see below; data in gray, model predictions in blue). As you can see, the predicted accuracy is quite good, especially once accuracy rises above chance levels.

Experiment 1

Experiment 2

Experiment 3

Experiment 4

We also performed formal model comparison to a simpler model that only allows the mean and SD of response preparation to vary as a function of the previous trial's accuracy (i.e. beta is identical after errors and after correct trials). We compared models based on their out-of-sample predictive fit via expected log pointwise predicted density (ELPD). We find that the model that allows beta to vary fits better in all four experiments. Although the relative likelihood of this model varies somewhat across the experiments (Exp 1: 10 times more likely, Exp 2: 2 times more likely, Exp 3: 13 times more likely, Exp 4: 4 times more likely), the pattern is quite clear. We have added this information to the methods section of the manuscript and include the above figure in the Supplemental information.

The fact that the effect resides in beta, can it be an artifact of the fact that at early PTs, there is simply no effect possible? Indeed, one is near floor level performance then. More generally, it would be nice to show via simulations (or perhaps earlier data) that effects can in principle reside in any of the parameters of the model.

Each of the parameters in the model uniquely affect a subject's conditional accuracy function (CAF) differently. The beta parameter is most sensitive to the asymptotic ceiling level of performance, which is where most of our behavioral results reside. (However note that the beta parameter contributes to accuracy at all time points where accuracy is above chance.) Although it is true that participants start out at chance levels of performance (before 200-300 ms), the other parameters of the model (mu and sigma) are most sensitive to the time after which accuracy rises above chance and the rate at which accuracy increases as PTs increase. Our model simulations in the bottom row of Figure 2 of the manuscript were intended to show how behavioral results would change if each parameter of the model were to change substantially following an error. To bolster this point, we have simulated data showing how slowing mu by 100 ms, increasing sigma by 50 ms, or both would affect the CAF (see plot below).

It is conceivable that a participant could produce any of these, but we simply did not observe these effects in our data. With regard to potential for effects to reside in other parameters, other work from our lab using this model has shown that increasing performance contingent rewards primarily affects the speed of cognitive processing (the mu parameter) albeit in a 2 alternative forced choice task rather than the 4AFC task used here (see experiment 3 in Adkins and Lee, PsyArxiv 2021; <https://doi.org/10.31234/osf.io/hv9mz>).

I find the distinction that is made in the model between “cognitive processing” (a normally distributed variable with a mean and a variance) and the efficacy of cognitive processing (beta), quite arbitrary. What warrants this distinction? Are there different brain areas assigned to these two aspects, or is it only a mathematical convenience?

We do not believe that this distinction is a simple mathematical convenience, though perhaps the specific term ‘efficacy’ is somewhat of an arbitrary choice. Data from the experiments presented here, some of our prior work (e.g. Adkins and Lee, 2021), and other research examining action slips all suggest that response errors can occur independently of the latency of response preparation. Our introduction of the term “efficacy” allows us to separately refer to the timing of processing independently from response execution. We have added to the introduction and discussion regarding the term ‘action slip’ (see below) to address this ambiguity in terms.

With regard to brain areas, we do not have the data to speak to this point. Given that action slips can occur for a variety of reasons (again, see below), we doubt that there is a single brain area responsible for ‘efficacy’.

Is there evidence that this type of model fits better than the more established type of sequential sampling (e.g., diffusion) model (see previous point). Of course, not everybody has to use diffusion models, but the fact that this type of model cannot even produce a positively skewed RT distribution, seems a big disadvantage to me. The rather subtle conceptual (and non-empirical) points that the authors make in this regard about deciding when to emit a response, are not really convincing to me. Relatedly, is the assumption that when cognitive processing is “ready”, a correct response is always given (but that this correct response can accidentally slip)?

While we believe it is beyond the scope of this paper to perform any direct comparisons between the modeling approach we take here and the various sequential sampling models, we have in pilot work examined how models such as the Drift Diffusion Model (DDM) can fit the data generated by our forced response paradigm. The DDM provides somewhat worse fits to our data, likely due to the fact that when PTs are very low, participants begin response preparation prior to the stimulus presentation (i.e. they are guessing). We have also attempted to use skewed distributions in our modeling approach in other preliminary work, but they do not provide better fits to our forced response data.

With regard to the point that people decide when to emit a response separately from the time it takes for response preparation, it has been shown empirically that all participants can make accurate responses much more quickly than their free RT would indicate by Haith et al (2016). We reference this work in the manuscript, but we did not discuss it fully for space considerations. In their work, Haith and colleagues showed that when forced to make a response, participants reach ceiling levels of accuracy at processing times that are faster than virtually their entire RT distributions. Essentially there is always some extra time that participants take before a response is made. Other related work by Wong and colleagues (2017) used a forced response paradigm to show that free RT can be heavily biased by a brief prior experience where faster RTs are necessary. These empirical papers show that RT is dependent on other factors aside from computation time. Even in an evidence accumulation framework, there is neurophysiological evidence from the EEG literature suggesting that sensory evidence accumulation as indexed by centro-parietal positivity is separable from preparatory motor activity as indexed by the lateralized readiness potential over motor cortex (e.g. Kelly and O’Connell, 2013). In the drift diffusion model, a very conservative response threshold would also have the function of choosing how long to accumulate evidence and approximately when to make a response (though the assumptions of the underlying processing are somewhat different). Adding a parameter to our model that controls this delay between computation and response could certainly produce skewed distributions seen in free RT.

However, we acknowledge that asserting that this represents a ‘decision’ about when to respond is one of several possible interpretations of prior work.

As far as the model is concerned, when the cognitive processing required to make a response is complete, the likelihood that the correct response is emitted is governed by the beta parameter. In the model, it is quite literally the likelihood of making the correct response given that it has been prepared at a given time point. In practice, it is possible that nothing is prepared on a small subset of trials and that would lead to a beta less than 1. However, we may have been too strong in wording the inferences that we draw from this. In addition to the action slip idea we originally advanced, we have outlined several other possibilities in the discussion section of the manuscript and why we think they may be unlikely in this case. These include: 1) A lack of perceptual processing on a subset of trials, 2) A failure in action selection, 3) Memory errors.

Line 186: If ... then these effects might disappear with a sufficiently long ITI. And the current theory does not “predict” that these effects might disappear? I do find the experiment informative, but I don’t see how it would disentangle the current theory from the orienting theory.

We now realize that we did devote sufficient space in the manuscript to contrasting our results with predictions from the orienting theory. Reviewer 3 had similar concerns, so we reworked the introduction and added language to the discussion to further expand on how our results relate to the orienting theory.

Also related to this point, the novel task that the authors present is actually a dual-task, because subjects must pay attention to the RT cue while waiting for / processing a stimulus. In this sense, the gain of not having the “when to respond” process (as the standard design presumably has), also comes with a cost of more difficult / less natural task.

We grant that participants have to monitor another stream of information to get the timing right in our forced response task. After training, participants get well entrained on the timing of when to make a response, but they are not perfect. This does make the task incrementally more challenging than a free RT version of the task. However, it is a bit unclear why this would fundamentally change the nature of post-error effects. We have added to the discussion to clarify this point.

One other disadvantage (or feature, depending on how looks at it), is that error probability is much higher than in standard designs. Thus, an error is also not so bad, or so unexpected as it typically is. It would be great if the authors can comment on this (or even better, present an analysis that addresses it).

While it is true that error probability is fairly high when PTs are very short, response accuracy when collapsing across all PTs is actually fairly high across all four experiments (~75%). While this accuracy rate is relatively low for simple 4AFC tasks like the one we employ here, many studies examining post-error employ conflict tasks (e.g. flanker, Simon, or Stroop tasks) or other tasks with similar levels of accuracy. For example, mean accuracy in Purcell and Kiani (2016) was 74.8% and mean accuracy across several experiments in Gjorgieva and Egner (2022) was ~78%.

It would also be interesting to see the effect of the type of previous error. For example, making

an error if one has 0 msec processing time available (i.e., on the previous trial), seems cognitively very different than if one had 1000 msec available (again, on the previous trial).

The authors might also consider exchanging error on trial $n - 1$ with error on trial $n + 1$. This is a standard type of procedure in the post-error slowing literature; indeed; if the trial $n - 1$ effect remains with $n + 1$ trials, it cannot be a causal effect (and is more likely due to a slow drift across trials). Of course, this is not a standard type of design (with much larger range of error rates than usual), but could still be of interest.

Preprocessing: I may have missed it, but what % of trials is removed by excluding responses outside the response window? Were subjects informed that they were too early/late in such case?

In experiments 1 and 2, participants were given timing feedback for 1000 ms after every trial (on time, too slow, or too fast). In experiments 3 and 4, we only gave timing feedback if they were too slow/fast and thus implicitly gave 'on time' feedback. This information was added to the Methods section of the manuscript.

We regret that we neglected to include the % of trials with timing errors. We have now added them to the manuscript and they are reproduced here.

Exp 1 36.9%

Exp 2 30.6%

Exp 3 40.3 %

Exp 4 35.8 %

We removed these trials from the analyses primarily for ease of interpretation given that often these timing errors skewed toward slower "late" responses (see figure below).

We were aiming to eliminate post-error slowing of responses to examine the effect of errors on other aspects of cognitive processing. Even though we always plot the accuracy in terms of the actual processing time (i.e. time from stimulus presentation to response, rather than stimulus to 'go' cue), including these trials would be a mix of slowing and other effects. With that said, including all trials regardless of timing does not change the results substantially (see figure below). Even at the longest processing times, there is a relatively static decrease in accuracy following errors that is indicative of a decrease in the β parameter.

Additionally, these timing errors do not seem to affect performance (timing accuracy nor response accuracy) on the subsequent trial (see figures below and response to reviewer 3 for more).

With regard to examining $n-1$ error trials, it is essentially similar to the $n+1$ analysis because they are essentially examining the same effect (i.e. do errors tend to follow one another). When examining post-error slowing in free RT paradigms, it is possible to examine $n-1$ and $n+1$ trials because there are two dependent measures (accuracy and RT). So you can look at RT on correct trials pre- and post-error. In our case, however, we only have accuracy. Nevertheless, we believe it is unlikely that this can be attributed to a slow drift in attention or brief periods of attentional lapses. One would expect that if a participant had a lapse in attention that spanned multiple trials (leading to a higher error rate) they would also be more likely to mistime their responses. In fact, the timing of the response is likely the feature of the task that requires the most vigilance. However, as shown in the plots below, it does not appear that timing errors follow one another or that timing errors tend to follow response errors.

Last sentence of abstract: PES may be A instead of B; A and B seem simply different formulations of the same concept to me. This makes me wonder what exactly has been shown by the paper. On line 92, a similar claim is made again. To clarify, the point of shifting along the speed accuracy curve is exactly to adapt (become slower but more accurate) in response to impairments in cognitive processing (here, after an error).

We regret the lack of precision here. What we were trying to get across is that post-error slowing may not simply reflect a shift in strategy (with no change in the underlying speed-accuracy tradeoff function). Instead, our data is showing that this SA tradeoff is fundamentally altered following an error and not just transiently in the first second or two after the error. We have edited the abstract to make this point clearer.

Line 42: "that are analyzed separately": Usually, a diffusion model (or similar evidence accumulation model is fit)

That is true. We have removed this phrase as it is not essential to the argument being made at that point in the introduction.

Line 71: Shouldn't alpha also be described as a parameter? Or is that fixed?

Yes, alpha is a free parameter in the model, but we did not allow this to vary as a function of accuracy on the previous trial. We reasoned that participants should be guessing randomly at the early time points when there isn't enough time given for processing of the stimulus. Indeed accuracy starts out at chance levels (25%) in all 4 experiments. We have stated plainly in the introduction that alpha is one of the free parameters in the model.

Reviewer #3 (Remarks to the Author):

The authors of this manuscript present the results of a model-based analysis of four behavioral online experiments addressing adaptive or maladaptive changes in information processing on trials following the commission of an error. They make use of a forced-response paradigm in which they manipulate the duration of stimulus processing prior to the forced response and the inter-trial interval (ITI). A response preparation model was built and free parameters indicating the speed and variance of response preparation and the efficacy of response execution were fit to the behavioral data. While response preparation was not significantly modulated after errors, efficacy of response execution was impaired and more perseverative slips were committed after errors. The authors suggest that these findings are not fully compatible with current theories of post-error adjustments and, in particular, post-error slowing.

The study addresses a research topic that has been debated for a long time and seems to add an interesting result to the literature. However, in my opinion the conceptual advancement provided by this manuscript is rather limited. The results do not seem to lead to a novel theoretical account that can explain the quite variable findings that have been reported previously based on a breadth of tasks. In other words, it remains unclear to me how the findings and conclusions can be extended to other tasks; they seem to be quite specific to the task studied here. Therefore, I believe that the results will not be of sufficient interest to the broad readership of this journal and might be better suited for more specialized journals. It also appears that the authors need to clarify the concepts and terms they use in the manuscript. I, furthermore, have a number of questions and concerns that are listed below:

We believe the value in our empirical work is not to advance a novel theoretical account that explains all post-error effects. Instead, we provide evidence across 4 experiments of a post-

error effect on behavior that is not easily described by the extant theories (i.e. response caution, orienting responses, or adaptive orienting). Our work also rules out some assumptions of these prior theories (i.e. our work shows that impairments are not transient and can still be observed several seconds following the commission of an error. Our hope is that we can move the field forward by providing new data that must be accounted for by any account of how cognitive processing is affected by errors. We are heartened that the reviewers remarked that our findings are "interesting", "innovative", and that they provide "important new insights" on the topic.

1. Abstract: I think it is not fair to state the dominant account of post-error slowing is to assume more cautious responses after errors. This view has been challenged multiple times and disentangling adaptive and maladaptive adjustments after errors has been the topic of many theoretical and empirical papers over the last 15 years or so. Some of these are, while being quite relevant, not mentioned in the manuscript, e.g., Wessel (2018) An adaptive orienting theory of error processing, *Psychophysiology*; Fischer et al. (2018) Cortical beta power reflects decision dynamics and uncovers multiple facets of post-error adaptation, *Nature Communications*; Steinhäuser et al. (2019) Rapid adaptive adjustments of selective attention following errors revealed by the time course of steady-state visual evoked potentials, *Neuroimage*. Taking together the current literature suggests at least to me that adaptive as well as maladaptive mechanisms kick in after an error and the relative timing and impact of these mechanisms depends on the type of task and contextual factors. Also the definition of "adaptive" or "maladaptive" heavily depends on the context and the goals of the acting person.

We thank the reviewer for pointing us toward more empirical and theoretical papers that regarding maladaptive adjustments post-error. We agree that errors clearly lead to both adaptive and maladaptive processing and we did not intend to claim that it has to be either/or. We have expanded our introduction and amended our abstract to more fully flesh out the orienting account up front in the manuscript rather than having it focused in the discussion. From our read of the literature, we maintain that the results we obtain here are not fully compatible with an adaptive orienting account and provide evidence for specific post-error deficits in processing that have not been reported previously. We have added language to the discussion to state this more explicitly in the manuscript.

2. The task is interesting and certainly adds a new perspective to the field. However, I am not sure whether it can be interpreted in such straightforward fashion as the authors suggest such that the "what" and "when" decisions could be unequivocally disentangled. It appears that participants have to do two tasks in parallel: (1) monitoring the square and pressing at the right time and (2) selecting the correct response. Thus, timing errors as well as action selection errors can occur and the analyses (as well as the model) do not address any potential interactions between both tasks. Other aspects such as response urgency (cf. work by Paul Cisek) is not taken into account. If the stimulus appears only briefly before the time point when the response should be executed, urgency (and response vigor) can be expected to very high. Again, multiple accounts for urgency have been suggested, e.g. an urgency signal, increases of gain in evidence accumulation and/or collapsing decision boundaries.

We are pleased the reviewer finds our work to be a fresh perspective! The timing component is an essential feature of our task and it is true that there is some dual-task load on our participants. However, we believe this load to be more minimal than the reviewer suggests. The

response timing required from trial to trial is identical (i.e. responses need to be emitted at exactly 2 seconds from the start of each trial). Participants get somewhat entrained to this timing and the timing cues aid in keeping this time, though they are most definitely not perfect.

The analyses we present in the manuscript are framed around key-press errors specifically and not timing errors. Key-press errors do not appear to affect timing accuracy on the next trial substantially across experiments (see figure below with the Bonferonni-corrected p-values from simple paired t-tests):

Additionally, timing errors have no effect on key-press accuracy on the subsequent trial (see figure below). It does not appear that there really is much of a post-timing-error effect, though this could be due the relatively high frequency of timing errors and the associated feedback. There does not appear to be an interaction here between timing errors and key-press errors.

The above analyses are now reported in the manuscript and these figures are included in a supplement.

With respect to response urgency, it is most definitely the case that participants know when to respond and they are instructed to make a random response at that time if they haven't had time to process the stimulus. It is actually our intention in the design to force this "urgency" upon them to interrogate the state of stimulus-response processing at each time point. Because we do not use an evidence accumulation framework here, we are agnostic as to whether these time deadlines increase the rate of evidence accumulation or abruptly collapse the decision threshold, although in some sense our modeling framework is more closely aligned with the latter. It assumes that stimulus-response processing takes some amount of time to complete (see the next response below) and when that occurs, the correct response is executed when initiated at the rate given by the beta parameter.

Also, is it valid to assume that participants guess randomly, "if stimulus-based response preparation is not yet finished" (p.3, l.34) What is meant by complete here? What, if some evidence is already accumulated, then the response will be biased and not random. Do participants prioritize responding in time over selecting the correct button with sufficient confidence?

We use the phrase "stimulus-based response preparation" to mean the processing necessary to produce the correct response (above and beyond chance level responding). In our view, this subsumes cases in which the accumulated evidence has biased the response toward the correct button press. Our data suggests that participants in all four experiments are guessing randomly up until they have had at least ~350 ms to prepare a response. In the framework we adopt here, that essentially means that "response preparation" is never finished until after this time-point. Although we do not adopt a strict evidence accumulation framework, this finding is consistent with the notion that it takes at least 350 ms of evidence accumulation to significantly bias responses toward the correct key (on a small subset of trials). Because we essentially do not allow participants to select their own evidence threshold, one can think of our design as

taking snapshots of the balance of the evidence that has been accumulated at each PT. This is similar to the assumptions made when using diffusion models to fit “interrogation” paradigms with response deadlines. Another way to think about what we mean from the perspective of an evidence accumulation framework is that if some evidence has accumulated at all above negligible levels and the motor processing associated with associated response has time to occur, then “stimulus-based response preparation” is complete. At early PTs, this doesn’t happen very frequently as evidenced by very low levels of accuracy at these time points. We have added to the discussion to make more explicit how our results could be interpreted from an evidence accumulation point of view.

3. What could be the mechanisms of action slips if response selection is complete?

As we note in the response to Reviewers 1 and 2 above, action slips could be due to several factors including: 1) A lack of perceptual processing on a subset of trials, 2) A failure in action selection, 3) Memory errors mapping the stimuli to the correct responses.

Our data here cannot definitely provide evidence for or rule out any of these possibilities, but we now discuss these possibilities more fully in the manuscript and allude to them when we introduce the term.

4. Important data is missing in the manuscript. The descriptive stats on error trial numbers and error rates, error types, timing errors etc. need to be reported. The trial numbers are important to assess the reliability and robustness of the analyses. How often did “too fast”/“too slow” feedback occur and how was their influence on subsequent trials taken into account? The methods section suggests that these trials were excluded. How many trials were excluded? Exclusion of trials also disrupts sequences of trials relevant for analysis (error-correct, correct-correct etc.). Was there a higher likelihood to make mistakes after timing feedback (too fast/slow)?

We regret this oversight. We have added this information to the manuscript (reproduced here).

Our analysis excluded trials where the current response (irrespective of the previous trial’s accuracy) was “too fast”/“too slow”. However, for each on-time trial we classified it as post-correct or post-error regardless of whether the previous response was well-timed or not. As the

reviewer is alluding to, this could be a problem if time errors lead to response errors, but this appears to be unequivocally not the case (see figure above).

5. If participants prioritize response timing over accuracy, it does not seem to be surprising that no post-error slowing was found. They just try to be in time at any cost. This urgency effect could counteract slowing. Post-error slowing has been suggested to result from motor inhibition. In connectionist models (Botvinick et al, 2001) as well as sequential sampling models (e.g., drift diffusion models) post-error slowing is implemented via a shift of the decision boundary. Purcell & Kiani (2016) as well as Fischer et al. (2018) provided empirical evidence supporting this account. It is not clear why a shift in the motor threshold would necessarily prolong "cognitive processing" or "preparation speed". And perhaps the threshold is not shifted in this task or any threshold shifts are compensated by a gain increase/an urgency signal.

We did not mean to imply that there is not a shift in decision boundary in free RT tasks. As the reviewer mentions, there are several papers providing empirical support for this idea (Dutilh et al 2012 also comes to this conclusion using the DDM). We grant that response caution occurs following an error. As we attempted to outline in the introduction however, response caution alone cannot explain the finding that responses are often reported to be slower AND less accurate following errors. This reduction in accuracy cannot be explained by a more cautious decision threshold. Orienting accounts suggest that dips in accuracy should be transient, but our results suggest that accuracy is reduced regardless of how much time is given following an error.

Our intention here was to actually use the forced response method to control for decision threshold in a sense. We believe our method essentially makes this an independent variable by forcing people to respond with predetermined amounts of processing time. (Although we grant at long PTs it is possible that a decision threshold has been crossed, they stop accumulating evidence even though the stimulus is still present on the screen, and a participant might simply be holding this response until the "go" signal.) Thus, we were attempting to get a clearer picture on what other changes in cognitive processing are occurring following errors that would lead to a decrease in accuracy.

Relatedly, some prior researchers have suggested that the inhibition that leads to post-error slowing might be more global than just motor inhibition and might extend into the inhibition of cognitive processing as well (e.g. Wessel and Aron, 2017). If true, this would suggest that we should observe a slowing of "cognitive processing"/"preparation speed". We have added language to the introduction to motivate this possible outcome of our experiments and have added to the discussion to expand on the implications of these findings.

It is indeed intriguing that error likelihood and perseveration is increased on post-error trials, but the authors do not seem to provide a theoretical account for that finding. What could lead to the effect that the previous response representation in the motor system is "overwritten" less efficiently after errors?

As we note in the responses above, it does not seem that this post-error perseveration can be accounted for by previous theories of post-error effects. However, we now outline some possibilities in the discussion (problems retrieving stimulus-response mappings, failure in

stimulus-based action selection, etc.).

6. p. 11, l. 22: "Instead, post-error slowing effects might be better explained by adaptive delays in response initiation to compensate for impairments stimulus-response processing that lead to more frequent action slips" - this is purely speculative and not supported by the data. If this speculation were true, what mechanism would trigger that additional adaptive delay? And how could it compensate for more frequent action slips?

We agree that this point is likely overly speculative and we have removed it from the manuscript.

REVIEWER COMMENTS

Reviewer #1 (Remarks to the Author):

The authors have addressed almost all of my concerns. I just have a few minor comments related to how the results are interpreted (or at least how this is presented and summarized in places). I think these are important to resolve to avoid the main findings of the paper being misconstrued.

In the abstract (and also lines 111-113, 303-305) the authors suggest that "post-error slowing may be an adaptive response to impaired cognitive processing that reflects a fundamentally altered relationship between the speed and accuracy of responses". It's not entirely clear how delaying of reaction times could be an adaptive response to a lowered "efficacy" (R3 has a previous comment on the last version of the paper). The results in fact imply that increasing RT will not be an effective means of compensating for whatever causes post-error effects, since the efficacy phenomenon is independent of RT. Potentially, the reduced accuracy makes it even more important to avoid errors arising from responding too quickly, so that RTs are increased to eliminate any risk of initiating before selection is complete? Though wouldn't this be "increased caution"? The interpretations and conclusions put forward feel slightly incongruous at times still, so I think it would be important for these critical points to be explained more clearly. In particular, the suggestion of post-error slowing being an "adaptive response" connotes the idea that post-error slowing is a compensation for slowed response preparation – inconsistent with what the results actually show. It's possible that both the phenomenon of perseveration after an error and the phenomenon of post-error slowing (if assessed in free RT conditions) could both be true but unrelated to one another.

line 336-338: "This suggests that post-error slowing and increased decision thresholds observed in prior work might not be compensating for current impairments in processing rather than simply just a reaction to negative feedback." Something wrong with how this sentence is written ("...might not... rather than...")

line 403: "efficacy was defined as the probability that a participant will express a response after it has been prepared". I find this statement a bit misleading. In truth, efficacy is a parameter of the model, and so it is awkward to also provide a definition beyond the scope of the model. There are a number of interpretations for how to interpret this parameter: e.g. as a failure to prepare the response in the first place, or as a failure to emit that prepared response.

line 445: "These results suggest a shift in the speed-accuracy tradeoff function" - I don't think this is an accurate way to summarize the results. 'shift' in this context normally refers to a shift along the speed axis, rather than the accuracy axis, and in any case it is more of a rescaling of that function than a 'shift'.

Reviewer #2 (Remarks to the Author):

I think the authors have responded very well to my comments, except for one thing: The $n+1$ analysis. I understand that in the standard post-error slowing analysis one has both accuracy and RT, and the $n+1$ analysis uses both. I also get that this is not exactly possible in this design. But the conceptual point still holds: Why would one not be able to carry out exactly the same effect as the authors have done, but analyse current-trial accuracy as a function of next-trial accuracy. and show that the beta parameter does not shift the way it does as a function of previous-trial accuracy.

I think it would be really convincing if the authors can show that results are not similar to their main analysis in this case. The analysis that the authors do provide is not uninteresting, but I think the proposed analysis would be much more convincing.

It's possible that I missed a conceptual point of why it is not doable (I apologise in such case); but I think it would be good if the authors can either explain convincingly why it is not possible or report the analysis (in supplement or main text, that is both fine for me).

Reviewer #2 (additional comments on your previous responses to Reviewer 3):

General comment: Reviewer 3 finds the task and theory limited to this experimental paradigm.

The authors note that they do not intend to describe a novel theory, but instead a behavioral effect (replicated across 4 experiments), in a novel experimental paradigm, that is hard to capture in current theories.

Although this is a very general comment (and thus for the authors also hard to rebut), I also see the point of reviewer 3. The task is a bit forced (when in daily life are we forced to respond (repeatedly) in a very limited time interval?), so it's also not obvious to what extent this novel paradigm brings problems to current theories. On the other hand, it is a novel perspective on task processing, extant experimental paradigms are also in some ways artificial (though less so, in my opinion, than the current one), and the lack of slowing in this task, demonstrated by their careful model analysis, is indeed surprising.

Specific comment 1: Abstract

This is well responded to by the authors.

Specific comment 2: Keypress and timing errors. This relates to general comment, that the task is a bit artificial. Specifically, the authors don't consider timing errors.

The authors provide a valid response demonstrating that timing errors are independent of keypress errors in this case. Although a good point, it remains open to what extent this generalizes to other (less artificial tasks). This seems more a question of "future study" to me than a strong limitation.

Specific comment 3: What is the mechanism of action for action slips?

The authors cannot really address that, but it would be unreasonable to expect a full explanation of that, so this is ok for me.

Specific comment 4: Missing data.

The authors now show (in a figure) the number of too slow or too fast responses. This is good, but also raises some issues. First, the % seems very high (hard to quantify with a figure, but I guess about 20%); how to handle them could definitely change the (modelling) results. Relatedly, the authors argue that time errors do not lead to response errors, but I can't see that from the graph, and so the issues remains open for me (apologies if I missed it).

Specific comment 5: If participants prioritize timing, it's not surprising that they show no processing time costs.

The authors mention that the paradigm is intended to control for decision threshold. They don't really address it, but then again it's hard to address the comment that a finding is "not surprising". I guess this all goes back to the same general comment: Is this novel paradigm + model a useful avenue to study post-error behaviour? I guess this depends on one's perspective to a large extent..

Specific comment 5b: What's the mechanism of action?

Again, the authors have no clear idea. It's not clear though if that is required; at least the observation itself seems quite robust. Whether this is truly interesting or "just" paradigm-dependent, will require

future work. It seems unreasonable to ask to disentangle that in a single paper.

Specific comment 6: Ok, they removed it.

In summary, they have been reasonable in their response, but the main comment "Is this a useful paradigm to study error and post-error processing?", remains, obviously. This is a matter of judgment. I can see the point of reviewer 3: The paradigm is artificial, and it forces a focus on response timing (so not surprising that the timing characteristics are different from other work). It's also odd to claim that free-response models and theories do not consider the "when" of responding. In fact, how to set the bound (the "when") has been topic of great interest in recent years (as another reviewer points out). I can also see the point of the authors that this may be a useful task to be used in future research and to be addressed by future theories of cognitive processing.

We would again like to thank the reviewers for their comments. Also, a special thanks to Reviewer #2 for additionally looking over our response to the third reviewer. We hope that we have satisfactorily responded to their comments below (reviewer comments italicized in blue, our response in black).

REVIEWER COMMENTS

Reviewer #1 (Remarks to the Author):

The authors have addressed almost all of my concerns. I just have a few minor comments related to how the results are interpreted (or at least how this is presented and summarized in places). I think these are important to resolve to avoid the main findings of the paper being misconstrued.

In the abstract (and also lines 111-113, 303-305) the authors suggest that "post-error slowing may be an adaptive response to impaired cognitive processing that reflects a fundamentally altered relationship between the speed and accuracy of responses". It's not entirely clear how delaying of reaction times could be an adaptive response to a lowered "efficacy" (R3 has a previous comment on the last version of the paper). The results in fact imply that increasing RT will not be an effective means of compensating for whatever causes post-error effects, since the efficacy phenomenon is independent of RT. Potentially, the reduced accuracy makes it even more important to avoid errors arising from responding too quickly, so that RTs are increased to eliminate any risk of initiating before selection is complete? Though wouldn't this be "increased caution"? The interpretations and conclusions put forward feel slightly incongruous at times still, so I think it would be important for these critical points to be explained more clearly. In particular, the suggestion of post-error slowing being an "adaptive response" connotes the idea that post-error slowing is a compensation for slowed response preparation – inconsistent with what the results actually show. It's possible that both the phenomenon of perseveration after an error and the phenomenon of post-error slowing (if assessed in free RT conditions) could both be true but unrelated to one another.

We again regret the lack of clarity in our writing on this point. We agree with the reviewer that our results show that lowered efficacy is independent of RT and that it is possible that the phenomenon of post-error slowing (in free RT scenarios) and perseveration could be unrelated to each other. The point we were attempting to make relies on the fact that although efficacy is lowered following an error, more time is still beneficial to performance. Imagine that a participant has a criterion that they would like to be accurate 85% of the time. There is some minimum preparation time needed to maintain this level of accuracy. Let us say that this value is 500 ms. Following an error, "efficacy" is reduced such that accuracy rate at 500 ms is now just 80% and they need a longer PT, say 550 ms, for an accuracy rate of 85%. In our task paradigm, participants can't do anything about this as they need to respond when we tell them to. However, in a free RT scenario if people have metacognitive awareness about their reduced efficacy they can choose to delay their responses to maintain a similar rate of accuracy. This does not mean that increasing RT can eliminate the post-error dip in efficacy. However, increasing RT can compensate for this dip by allowing a person to maintain a similar level of

accuracy in post-correct and post-error trials. This would be “increased caution” as the reviewer points out, but it’s only possible in a free RT scenario and not in our forced response scenario.

We have amended to abstract and the discussion to make it clear that we are referring to post-error slowing in free RT tasks and to emphasize that it is possible that what we observe here could be an new independent effect of errors.

line 336-338: "This suggests that post-error slowing and increased decision thresholds observed in prior work might not be compensating for current impairments in processing rather than simply just a reaction to negative feedback." Something wrong with how this sentence is written ("...might not... rather than...")

Fixed. It should be ‘might be’ rather than ‘might not be’.

line 403: "efficacy was defined as the probability that a participant will express a response after it has been prepared". I find this statement a bit misleading. In truth, efficacy is a parameter of the model, and so it is awkward to also provide a definition beyond the scope of the model. There are a number of interpretations for how to interpret this parameter: e.g. as a failure to prepare the response in the first place, or as a failure to emit that prepared response.

We have changed this line to more explicitly state that this is a parameter in the model and not necessarily how one needs to interpret the resulting change in this parameter:

“In the model, ‘efficacy’ is the parameter that controls the probability that a participant will express a response after it has been prepared. Although there are several ways to interpret this parameter, a natural psychological interpretation of efficacy is participants’ confidence in their selected response.”

line 445: "These results suggest a shift in the speed-accuracy tradeoff function" - I don't think this is an accurate way to summarize the results. 'shift' in this context normally refers to a shift along the speed axis, rather than the accuracy axis, and in any case it is more of a rescaling of that function than a 'shift'.

We have replaced this to say a “change in the shape of the speed-accuracy tradeoff function” to make it clear that we aren’t talking about a shift in the speed axis.

Reviewer #2 (Remarks to the Author):

I think the authors have responded very well to my comments, except for one thing: The n+1 analysis. I understand that in the standard post-error slowing analysis one has both accuracy and RT, and the n+1 analysis uses both. I also get that this is not exactly possible in this design. But the conceptual point still holds: Why would one not be able to carry out exactly the same effect as the authors have done, but analyse current-trial accuracy as a function of next-trial accuracy. and show that the beta parameter does not shift the way it does as a function of previous-trial accuracy.

I think it would be really convincing if the authors can show that results are not similar to their main analysis in this case. The analysis that the authors do provide is not uninteresting, but I

think the proposed analysis would be much more convincing.

It's possible that I missed a conceptual point of why it is not doable (I apologise in such case); but I think it would be good if the authors can either explain convincingly why it is not possible or report the analysis (in supplement or main text, that is both fine for me).

The issue with this analysis is that it is not independent from the post-error effect that we report. In fact, it is guaranteed that they will look similar. We know from our post-error analysis that errors tend to follow one another. At every PT, an error is more likely following another error. But that also means that trials before an error are more likely to be an error! Both analyses essentially describe the same effect: errors cluster together temporally across trials. As we noted in our previous response, in free RT tasks it is possible to look at n-1 and n+1 trials because there is another dependent variable (RT) on correct trials to examine pre- and post-error. In our case, we know that n+1 is more likely to be an error, but that means that n-1 is also likely to be an error. We believe what the reviewer is trying to get at is that it is important to know if there is some change in state preceding the first error that describes the results seen here such that the change in efficacy is not a result of the error per se, but rather this change in state. We are hopeful that the analysis that we presented shows that this is somewhat unlikely in that attentional lapses would likely lead to timing errors as well as response errors, but we don't observe any effect there.

Reviewer #2 (additional comments on your previous responses to Reviewer 3):

General comment: Reviewer 3 finds the task and theory limited to this experimental paradigm.

The authors note that they do not intend to describe a novel theory, but instead a behavioral effect (replicated across 4 experiments), in a novel experimental paradigm, that is hard to capture in current theories.

Although this is a very general comment (and thus for the authors also hard to rebut), I also see the point of reviewer 3. The task is a bit forced (when in daily life are we forced to respond (repeatedly) in a very limited time interval?), so it's also not obvious to what extent this novel paradigm brings problems to current theories. On the other hand, it is a novel perspective on task processing, extant experimental paradigms are also in some ways artificial (though less so, in my opinion, than the current one), and the lack of slowing in this task, demonstrated by their careful model analysis, is indeed surprising.

We'd like to again thank the reviewer for going through these responses! It appears there is only one specific comment that requires a response from us (see below)

Specific comment 4: Missing data.

The authors now show (in a figure) the number of too slow or too fast responses. This is good, but also raises some issues. First, the % seems very high (hard to quantify with a figure, but I guess about 20%); how to handle them could definitely change the (modelling) results.

Relatedly, the authors argue that time errors do not lead to response errors, but I can't see that from the graph, and so the issues remains open for me (apologies if I missed it).

We acknowledge that the % of timing errors is relatively high and is an unfortunate (but necessary!) feature of the experimental design. We have reproduced the plot here showing that timing errors don't lead to response errors. In each experiment, accuracy rate (response errors) appear identical regardless of whether the previous trial was a timing error (mistimed) or not (on-time).

REVIEWERS' COMMENTS

Reviewer #1 (Remarks to the Author):

The authors have addressed all of my concerns.

Reviewer #2 (Remarks to the Author):

I want to comment on just the $n+1$ analysis. I agree with the authors that in this case, because the independent and the dependent variable both refer to the same variable (accuracy), the $n-1$ and $n+1$ analysis would yield the same result. (I apologise for not thinking sufficiently deeply about it in the first round.)

But it also means, as the authors also mention, that the effect simply means that errors are clustered temporarily. And instead of a strength, this is a weakness of the approach. Indeed, the title suggests that this approach allows determining what happens AFTER an error; whereas it can only determine what happens in the surrounding of an error. Errors are clustered in time, perhaps because errors distract in the specific way that the authors propose, or perhaps simply because there are periods of more vs less distraction in the experiment.

This is different in experiments where RT is the dependent variable (as the authors also acknowledge). I am fine with the current analysis, but I think it's worthwhile to point out the difference in the discussion section.

We are thankful for the reviewers comments and we are glad we have sufficiently responded to Reviewer #1's concerns. Our responses to Reviewer 2's comments are below in blue.

Reviewer #2 (Remarks to the Author):

I want to comment on just the n+1 analysis. I agree with the authors that in this case, because the independent and the dependent variable both refer to the same variable (accuracy), the n-1 and n+1 analysis would yield the same result. (I apologise for not thinking sufficiently deeply about it in the first round.)

But it also means, as the authors also mention, that the effect simply means that errors are clustered temporarily. And instead of a strength, this is a weakness of the approach. Indeed, the title suggests that this approach allows determining what happens AFTER an error; whereas it can only determine what happens in the surrounding of an error. Errors are clustered in time, perhaps because errors distract in the specific way that the authors propose, or perhaps simply because there are periods of more vs less distraction in the experiment.

This is different in experiments where RT is the dependent variable (as the authors also acknowledge). I am fine with the current analysis, but I think it's worthwhile to point out the difference in the discussion section

We agree with the reviewer that our results show that errors cluster together in time. Although our control analyses examining timing errors suggest that errors and the post-error effects we report are likely not driven by lapses in attention (i.e. participants' timing of responses are quite accurate), we cannot entirely rule out this account. We have added a paragraph to the discussion to address the nuance on this point. We reproduced this passage below:

One standard control analysis used in free RT investigations of post-error effects is to examine RT on trials preceding errors¹. The upshot of this type of analysis is to ensure that any slowing of RT that is observed following an error is due to the commission of an error and not simply a result of lapses in attention where clusters of trials have both slower RTs and reduced accuracy. That is, if RT on correct trials preceding errors is also slower than the average RT of correct trials across the entire experiment, any post-error slowing that is observed is less likely to be due to the error itself. Most studies, however, show that RT on trials preceding errors is faster than the average correct response²⁵. One shortcoming of the forced-response approach we adopt here is that we cannot perform an analogue of this control analysis that independently examines trials pre- and post-error. As we only have a single dependent variable of interest (accuracy), once we uncover that trials following an error (N+1) are more likely to display reduced accuracy and that errors tend to follow one another, it is necessarily true that trials preceding an error (N-1) are also more likely to display reduced accuracy. Thus, looking at accuracy alone makes it more difficult to rule out the notion that the results presented here are due to lapses in attentiveness that span multiple trials. However, if the reduced accuracy rate we observe following errors were due multi-trial

periods of inattentiveness, we would also expect to observe other deficits in performance. For example, more errors and variability in the timing of responses. We did not find such an effect. Key-press errors and timing errors appear to be independent in our data and we did not observe a higher likelihood of timing errors on trials adjacent to key-press errors. This gives us more confidence that the increase slips of action we observe following errors are a result of post-error processing, though we cannot completely rule out that there may be some other process underlying the serial dependence we report here.